Citation: *Molecular Systems Biology* 9:705
www.molecularsystemsbiology.com

# Perturbation of the mutated EGFR interactome identifies vulnerabilities and resistance mechanisms

Jiannong Li[1], Keiryn Bennett[2], Alexey Stukalov[2], Bin Fang[3], Guolin Zhang[1], Takeshi Yoshida[4], Isamu Okamoto[4], Jae-Young Kim[1], Lanxi Song[1], Yun Bai[1], Xiaoning Qian[5], Bhupendra Rawal[6], Michael Schell[6], Florian Grebien[2], Georg Winter[2], Uwe Rix[7], Steven Eschrich[8], Jacques Colinge[2], John Koomen[3], Giulio Superti-Furga[2,*] and Eric B Haura[1,*]

[1] Department of Thoracic Oncology, H. Lee Moffitt Cancer Center and Research Institute, Tampa, FL, USA, [2] CeMM Research Center for Molecular Medicine of the Austrian Academy of Sciences, Vienna, Austria, [3] Proteomics and Molecular Oncology Program, H. Lee Moffitt Cancer Center and Research Institute, Tampa, FL, USA, [4] Center for Clinical and Translational Research, Kyushu University Hospital, Fukuoka, Japan, [5] Department of Computer Science and Engineering, University of South Florida, Tampa, FL, USA, [6] Biostatistics Departments, H. Lee Moffitt Cancer Center and Research Institute, Tampa, FL, USA, [7] Drug Discovery Department, H. Lee Moffitt Cancer Center and Research Institute, Tampa, FL, USA and [8] Department of Biostatistics and Bioinformatics, H. Lee Moffitt Cancer Center and Research Institute, Tampa, FL, USA
* Corresponding authors. G Superti-Furga, CeMM Research Center for Molecular Medicine of the Austrian Academy of Sciences, Lazarettgasse 14, AKH BT25.2, 1090 Vienna, Austria. Tel.: + 43 1 40 160 70001; Fax: + 43 1 40 160 970000; E-mail: gsuperti@cemm.oeaw.ac.at or EB Haura, Department of Thoracic Oncology, Chemical Biology and Molecular Medicine Program, H. Lee Moffitt Cancer Center and Research Institute, MRC3 East, Room 3056F, 12902 Magnolia Drive, Tampa, FL 33612-9497, USA. Tel.: + 1 813 903 6827; Fax: + 1 813 903 6817; E-mail: eric.haura@moffitt.org

We hypothesized that elucidating the interactome of epidermal growth factor receptor (EGFR) forms that are mutated in lung cancer, *via* global analysis of protein–protein interactions, phosphorylation, and systematically perturbing the ensuing network nodes, should offer a new, more systems-level perspective of the molecular etiology. Here, we describe an EGFR interactome of 263 proteins and offer a 14-protein core network critical to the viability of multiple EGFR-mutated lung cancer cells. Cells with acquired resistance to EGFR tyrosine kinase inhibitors (TKIs) had differential dependence of the core network proteins based on the underlying molecular mechanisms of resistance. Of the 14 proteins, 9 are shown to be specifically associated with survival of EGFR-mutated lung cancer cell lines. This included EGFR, GRB2, MK12, SHC1, ARAF, CD11B, ARHG5, GLU2B, and CD11A. With the use of a drug network associated with the core network proteins, we identified two compounds, midostaurin and lestaurtinib, that could overcome drug resistance through direct EGFR inhibition when combined with erlotinib. Our results, enabled by interactome mapping, suggest new targets and combination therapies that could circumvent EGFR TKI resistance.
*Molecular Systems Biology* **9**: 705; published online 5 November 2013; doi:10.1038/msb.2013.61
*Subject Categories:* proteomics; signal transduction
*Keywords:* epidermal growth factor receptor; interactome; lung cancer; proteomics; tyrosine kinase inhibitor

## Introduction

Somatic mutations in the epidermal growth factor receptor (EGFR) result in enhanced receptor signaling, predicting sensitivity to EGFR tyrosine kinase inhibitors (TKIs), such as erlotinib and gefitinib, in subsets of patients with lung adenocarcinoma (Lynch *et al*, 2004; Paez *et al*, 2004; Pao *et al*, 2004). Both deletions in exon 19 and a point mutation that substitutes an arginine for a leucine at codon 858 (L858R) in exon 21 are the most common *EGFR* mutations (Murray *et al*, 2008; Rosell *et al*, 2009; Tanaka *et al*, 2010; Yoshida *et al*, 2010). These common mutant EGFR proteins lead to constitutive activation of downstream extracellular signal-regulated kinase (ERK), phosphoinositide 3-kinase (PI3K)/Akt, and STAT signaling, resulting in 'oncogene addiction' and tumor cell growth and survival (Sordella *et al*, 2004). Nonetheless,

mechanisms, such as gain of a secondary 'gatekeeper' mutation in EGFR (T790M), MET gene amplification, and epithelial–mesenchymal transition, can rapidly lead to drug resistance and limit the curative potential of EGFR TKIs (Pao *et al*, 2005; Bean *et al*, 2007; Engelman *et al*, 2007; Sequist *et al*, 2011; Suda *et al*, 2011). Approaches to overcoming resistance include use of irreversible EGFR inhibitors, agents directed specifically against T790M variants, heat-shock protein 90 (HSP90) inhibitors to prevent EGFR maturation, combined EGFR and MET inhibition, and dual MEK/PI3K inhibition (Shimamura *et al*, 2008; Faber *et al*, 2009; Zhou *et al*, 2009; Sequist *et al*, 2010a, b). However, to date, patients cannot effectively overcome resistance; thus, this remains an ongoing treatment dilemma.

We hypothesized that an interactome-based view of mutated EGFR in disease-relevant cells could produce insight

into how survival signals are transduced and could lead to new therapeutic targets and strategies to overcome resistance to EGFR TKI. Critical to protein function and signaling is the formation of complexes and networks of proteins that act in concert to produce a physiological signal. State-of-the-art mass spectrometry can now accurately map protein–protein interaction complexes and larger scale protein–protein interaction networks or interactomes (Gavin *et al*, 2002; Henney and Superti-Furga, 2008; Glatter *et al*, 2009; Gstaiger and Aebersold, 2009; Li *et al*, 2010). Interactomes can harbor subnetworks important in transducing signals from upstream cancer drivers; thus, examining interactomes would allow a better understanding of proteins involved in drug sensitivity or resistance (Astsaturov *et al*, 2010). In this study, we produced an EGFR interactome that itself can be viewed as a target for therapy, as opposed to single gene-based targeting strategies. Our integrative approach combined mass spectrometry-based interactome mapping with RNA interference functional analysis to gain insight into the survival machine produced by mutant forms of EGFR. To accomplish this goal, we experimentally derived a mutant EGFR interactome using disease-specific EGFR isoforms directly in lung cancer cells harboring EGFR mutations and hypersensitive to EGFR inhibitors using tandem affinity purification–liquid chromatography–mass spectrometry (TAP-LC-MS/MS) (Figure 1). We also directly examined proteins in complex with mutant EGFR proteins compared to wild-type EGFR proteins in immortalized epithelial cells using TAP. Using these results, along with secondary TAP experiments, we produced a mutant EGFR interactome by combining protein–protein interaction data along with phosphotyrosine proteomics data. The resulting mutant EGFR interactome reference map was used to functionally interrogate targets in EGFR-mutant lung cancer cell lines, leading to identification of new targets important for EGFR-driven survival. Lastly, we searched drug-target databases to identify compounds reported to target key network proteins and identify two compounds with gatekeeper EGFR mutation effects that showed combined effects with erlotinib in drug-resistant cell lines.

## Results

### A mutant EGFR interactome in disease-relevant lung cancer cell lines

Exon 19 deletion E746-A750 mutant forms of EGFR were used as a bait for TAP-MS experiments in lung cancer cells harboring mutant EGFR proteins and highly sensitive to EGFR TKI. Because the EGFR interactome is highly dependent on cell context, this approach offers the best chance of recovering pertinent proteins and mechanisms, a potential flaw in using cells other than lung cancer cells driven by mutant EGFR. We constructed Strep-HA-tagged versions of exon 19 deletion E746-A750 EGFR, which were then expressed in lung cancer cells using retroviruses. Retroviral expression is preferred to avoid highly overexpressed bait proteins that could lead to spurious interactions. We compared protein expression levels of the exogenously tagged bait proteins to endogenous proteins for these and subsequent experiments below and found nearly equal amounts of each protein type (Supplementary Figure S1A). A two-step process of biochemical purification (TAP) coupled to LC-MS/MS was

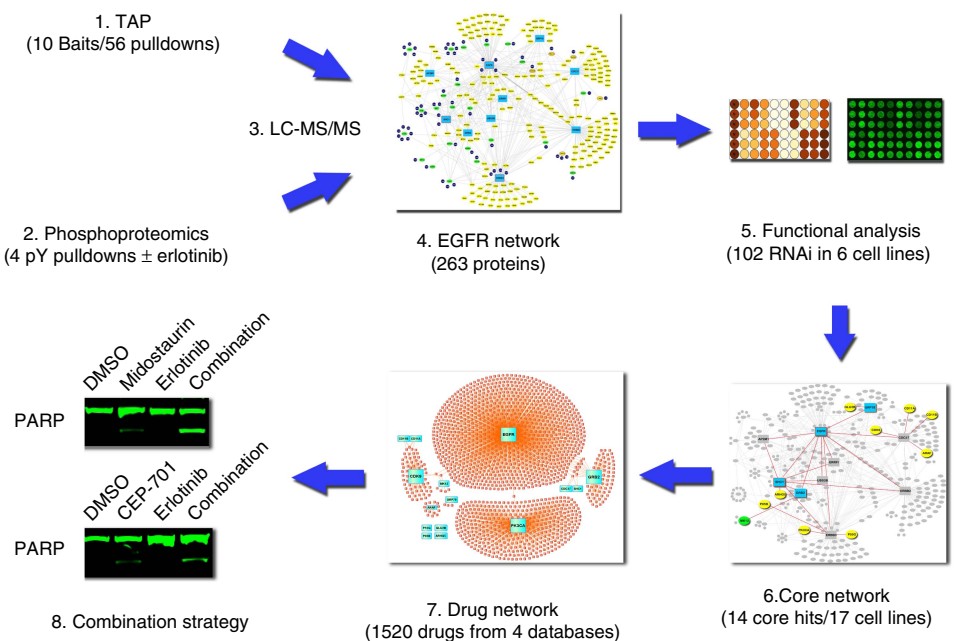

**Figure 1** Workflow. A physical protein–protein interaction network or interactome was experimentally derived using tandem affinity purification (1) and phosphotyrosine (pY) proteomics (2) in conjunction with liquid chromatography–mass spectrometry (LC-MS/MS) (3) centered on somatically mutated and drug-sensitive forms of EGFR in lung cancer cell lines (4). The interactome was perturbed using RNA interference (5) to identify a core EGFR network (6) characterized by proteins required for maintenance of cell viability across multiple ($N = 17$) lung cancer cell lines, including cell lines harboring EGFR mutation and dependent on EGFR ($N = 5$), and those that have wild-type EGFR and lack dependence on EGFR for survival ($N = 12$). Rationale combinations of drugs were informed by a curated drug network (7) linked to key survival nodes within the interactome that were validated in TKI-resistant lung cancer cell lines (8).

subsequently used to purify protein complexes (Haura *et al*, 2011). We performed EGFR TAP in two lung cancer cell lines: PC9 cells, which harbor an exon 19 deletion E746-A750 EGFR mutation and contain five copies of the *EGFR* gene per cell, and HCC827 cells, which harbor an exon 19 deletion E746-A750 EGFR mutation and contain 35 copies of the *EGFR* gene per cell (Soh *et al*, 2009). In addition to using EGFR as bait, we also expressed a tagged version of ERBB3 because of its previously demonstrated role in mutant EGFR signaling (Engelman *et al*, 2005). Finally, to gauge non-specific proteins bound to bait proteins and enrichment of proteins in EGFR, we expressed a Strep-HA-tagged green fluorescent protein (GFP) in the same lung cancer cells, allowing an aggregate of four GFP control pulldowns to be subsequently compared with mutant EGFR or other bait proteins.

After subtraction of proteins found in the GFP pulldown, we identified 24 putative interacting proteins of EGFR in the PC9 cells and 10 putative interacting proteins in the HCC827 cells (Figure 2A; Supplementary Figure S1B; Supplementary Data File S1). These findings included adaptors involved in receptor tyrosine kinase (RTK) signaling (GRB2, SHC1), the negative feedback inhibitor of EGFR signaling ERRFI (ERBB receptor feedback inhibitor 1), also known as MIG-6, and UBS3B, also known as STS-1 (suppressor of T-cell receptor signaling 1). STS-1 has been found to protect RTKs, such as EGFR and PDGFR, from lysosomal degradation, whereas the phospho-glycerate mutase of STS-1 has been shown to dephosphorylate wild-type EGFR at multiple tyrosines, leading to signaling termination and halting of endocytosis (Kowanetz *et al*, 2004; Raguz *et al*, 2007). We also observed four chaperones or co-chaperones (GRP78, HS90A, HS90B, and CDC37), other RTKs (ERBB2), tubulin proteins (TBA4A), and proteins known to be involved in RTK trafficking (AP2M1, AP2A1, AP2A2, and AP2B1). We validated a set of these observations by immunoprecipitation and western blotting (Supplementary Figure S1B). Although heat-shock proteins could be considered as abundant contaminants in TAP experiments, HSP90 proteins are known to be involved in mutant EGFR signaling, and we also identified the kinase chaperone and HSP90 co-chaperone CDC37, suggesting specific functional roles for these and possibly other chaperones in the EGFR and ERBB3 complexes (Shimamura *et al*, 2005). We identified 27 putative interacting proteins of ERBB3 in the PC9 cells and 38 putative interacting proteins in the HCC827 cells (Supplementary Figure S1B; Supplementary Data File S1), including multiple members of PI3K (PK3CA, PK3CB, P85A, P85B, and P55G) and multiple heat-shock proteins involved in protein chaperones (GRP78, HS90A, HS90B, and CDC37). ERBB3 also co-purified EGFR and the adaptor proteins GRB2 and SHC1.

We performed additional TAP coupled with LC-MS/MS experiments to directly compare EGFR binding proteins between mutant and wild-type forms of EGFR. We hypothesized that this approach would better define interacting proteins related to driver EGFR mutations as opposed to wild-type EGFR. We generated immortalized human bronchial epithelial cells (AALE) stably expressing Strep-HA tagged wild-type EGFR or exon 19 deletion E746-A750 EGFR mutation along with control-tagged GFP bait. The use of isogenic cell lines could characterize complexes associated with disease mutations since baits are expressed in identical proteomes. We

observed both a larger set of prey proteins in wild-type EGFR ($N = 57$) than in mutant EGFR ($N = 30$) and different sets of prey proteins associated with the different EGFR isoforms, despite equal if not more EGFR bait identified in the MS runs (Figure 2B and lists of prey proteins in Supplementary Data File S1). Mutant EGFR pulldowns identified MAPK adaptors GRB2 and SHC1 as well as ERRFI and UBS3B tyrosine phosphatase, all consistent with our results described in the lung cancer cell lines harboring mutant EGFR. Wild-type EGFR isoforms, on the other hand, were associated with more abundant levels of a larger set of transporter proteins including multiple adaptor protein complex (AP-2) isoforms, and vacuolar protein sorting-associated protein 13A (VP13A gene), which is known to be involved in receptor-mediated endocytosis and vesicle trafficking. Wild-type EGFR complexes were also enriched for a set of catenin proteins (a1, a2, b1, and d1) along with a known interacting protein ARMC8 and integrin-b4, suggesting a possible role in directing adherin junctional signaling. We examined whether prey proteins found from wild-type EGFR were part of known complexes (Figure 2C). We identified one sub-complex consisting of proteins involved in downregulating signaling from RTKs, including catenin and cadherin proteins along with tyrosine phosphatases PTPRD and PTPRF. A second sub-complex indicates proteins among the AP-2 subunits. From these results, we conclude that disease-relevant EGFR isoforms produce unique protein complexes characterizing an activation state leading to MAPK engagement as opposed to an inactivator state characterized by complexes associated with receptor trafficking.

We next focused our attention on a second round of TAP, to walk the interactome by using proteins predominately found in lung cancer cell line and AALE TAP data as secondary baits, with a goal to build a larger interactome centered on mutant EGFR. We produced Strep-HA tagged versions of GRB2, SHC1, ERRFI, UBS3B, ERBB2, GRP78, AP2M1, and CDC37. GRB2, SHC1, ERRFI, and UBS3B were found both in EGFR-mutant lung cancer cell lines and in AALE cells with mutant EGFR as a bait and thus represented logical choices to further examine as secondary baits. In addition, GRB2 and SHC1 were chosen based on their roles in linking EGFR to downstream MAPK signaling, while ERRFI and UBS3B were chosen based on the previous studies suggesting their roles in mediating RTK signaling. ERBB2, which was found in EGFR complexes in the lung cancer cell lines, and in complex with wild-type EGFR in the AALE system, was chosen based on its potential as an important heterodimeric protein with EGFR and ERBB3 and its role in EGFR TKI resistance (Takezawa *et al*, 2012), GRP78 (found in lung cancer cell lines and in both mutant and wild-type EGFR complexes in AALE) and CDC37 (found in lung cancer cell lines but not in AALE) were chosen as chaperones with potential functional importance. Finally, AP2M1, found in lung cancer cell line TAP and also identified with wild-type EGFR in AALE cells, was chosen as a representative member of adapter proteins important in EGFR endocytosis (Huang *et al*, 2004).

One possible limitation of this group of experiments is that TAP-MS may identify only stable and/or strong interactions between two proteins and may miss weak yet important interactions (such as pTyr-SH2) that are unable to survive the two-step biochemical purification with TAP. To address this limitation, we simultaneously performed tyrosine

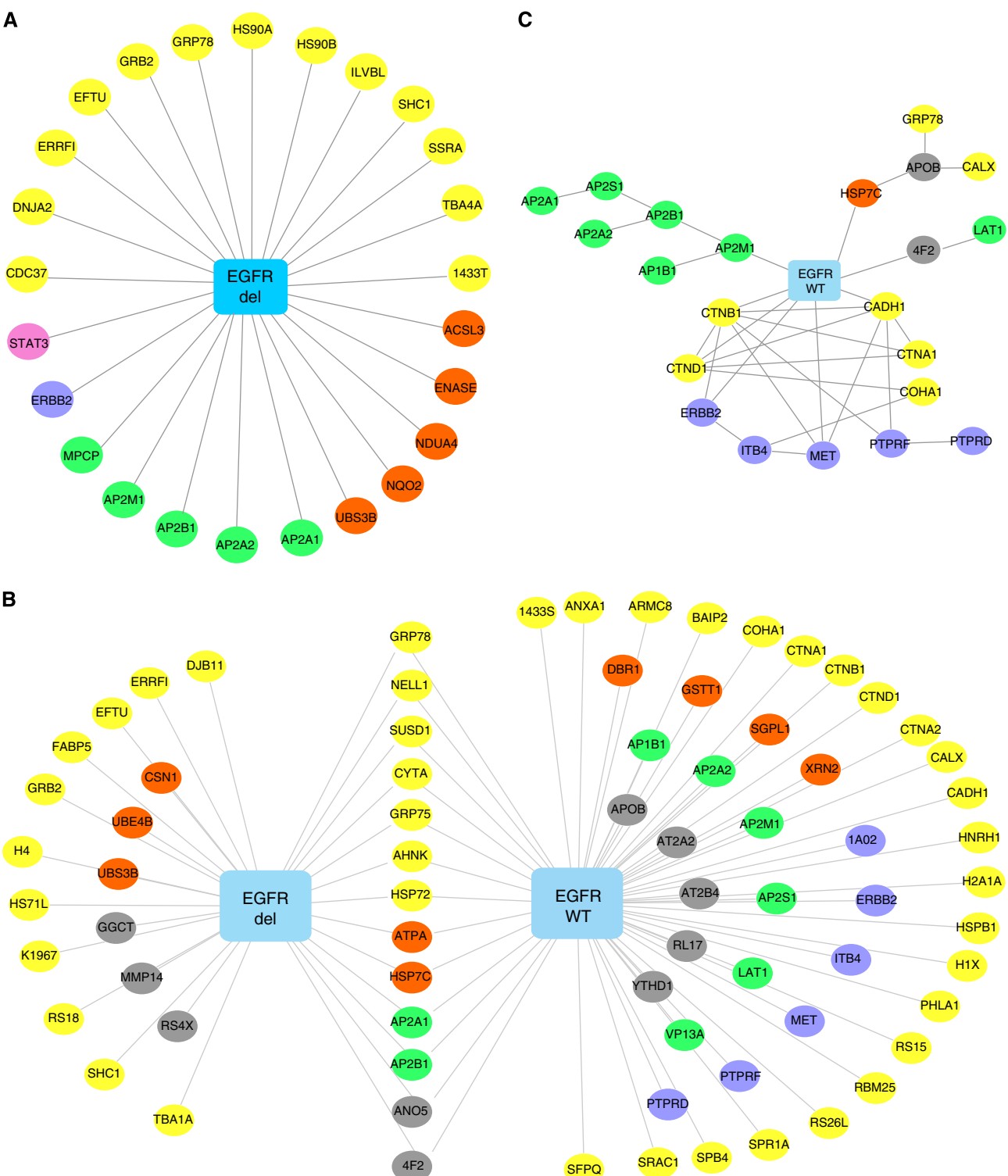

**Figure 2** Mutant EGFR complexes in lung cancer cell lines and immortalized human bronchial epithelial cells. EGFR protein complexes were identified by TAP. Protein complexes were visualized by Cytoscape (http://cytoscape.org/). The blue rectangle nodes represent the bait proteins and ellipse nodes represent prey proteins. The identified proteins were annotated and classified by its biological functions *via* MetaCore (http://portal.genego.com) (MetaCore, CA) and colored according to the functional groups: binding protein (yellow), enzyme (orange), transporter (green), receptor (purple), transcription factor (pink), and others (gray). (**A**) EGFR interaction in PC9 and HCC827 cell lines stably expressing exon 19 deletion E746-A750 EGFR mutation. (**B**) EGFR interaction in immortalized bronchial epithelial cell lines (AALE) cell lines stably expressing Strep-HA tagged wild-type EGFR or exon 19 deletion E746-A750 EGFR mutation; (**C**) Sub-complexes in AALE cell lines stably expressing wild-type EGFR were constructed in Cytoscape based on the BisoGenet data-mining search.

phosphoproteomics in PC9 cells (Rush *et al*, 2005; Li *et al*, 2010), and using protein–protein interaction databases, we linked pTyr-containing proteins to proteins identified in our TAP-MS experiments. Our previous study using purified SH2 domains to characterize phosphotyrosine signaling in lung cancer cells found PC9 clustering among other EGFR-mutant cells (Machida *et al*, 2010). Similar results were found in a previous study using anti-phosphotyrosine antibodies and mass spectrometry (Rikova, *et al*, 2007). On the basis of these results, we felt single analysis was a reasonable approach, especially as we were sampling to gain additional proteins to insert into the interactome and large number were already identified using TAP. Phosphopeptide immunoprecipitation and purification were performed in PC9 cells treated with 1 μM of erlotinib or DMSO as a control for 1 h with results analyzed by nano-LC-MS/MS. We identified 368 unique phosphotyrosine phosphopeptides corresponding to 186 unique proteins. For each detected phosphorylation site, the numbers of spectra that contain the modification as well as the non-phosphorylated ones were calculated separately for the group of samples treated with erlotinib and the DMSO control samples. For both phosphorylated and non-phosphorylated site states, the Pearson's chi-squared goodness-of-fit test *P*-value was calculated to estimate the significance of difference in the spectral counts of given state between the treated and non-treated samples. The two *P*-values were multiplied and the cutoff of 0.1 was applied to the resulting score to select the sites with significant changes in phosphorylation upon erlotinib treatment. We found 66 unique phosphotyrosine sites corresponding to 35 unique proteins were significantly perturbed by erlotinib in PC9 cells. The TAP interaction network was then extended by adding known protein–protein associations between the proteins already present in the TAP network and the ones containing phosphorylated tyrosine sites significantly altered by erlotinib. The list of candidate interactions was obtained from an internal database that aggregates public protein–protein interactions deposited to IntAct, BioGrid, MINT, DIP, HPRD, and InnateDB.

Rewardingly, the individual complexes collectively yielded a coherent interactome, consisting of 263 different proteins (Figure 3A; Supplementary Data files S1 and S2). Of the 263 proteins, 240 were contributed by TAP experiments while 23 pTyr proteins (shown with green ellipse nodes) were added to the EGFR network based on above database searching (Supplementary Figure S6). GRB2 surprisingly co-purified only a few proteins, including EGFR, ARHG5, UBS3B, ERRFI, and SHC1. Our results suggest that GRB2 may play a rather focused role in promoting associations with only selected effector pathways. SHC1 co-purified EGFR, GRB2, ARHG5, PTPN12, AP2A1 and AP2M1, ERFFI, and UBS3B. PTPN12, also known as PTP-PEST, is a protein tyrosine phosphatase that has been reported to dephosphorylate SHC1, Pyk2, Fak, and Cas and to inactivate the Ras pathway (Veillette *et al*, 2009). PTPN12 has recently been shown to be downregulated in some cancers, leading to hyperactivity of RTKs (EGFR and MET) and hypersensitivity to TKIs that inhibit these kinases (Sun *et al*, 2011). We found that ARHG5, a guanine exchange factor implicated in activation of the Rho pathway, was associated with GRB2 and SHC1. UBS3B predominately bound to EGFR with more limited peptide counts for GRB2 and ERFFI. ERFFI

could co-purify EGFR, GRB2, UBS3B, ERBB2, and multiple heat-shock proteins.

We found PI3K subunits to be exclusively in complex with ERBB3 but not with EGFR or ERBB2. This is consistent with previous studies showing that ERBB3 is the preferential partner for PI3K binding among the HER family of proteins (Engelman *et al*, 2005). ERBB3 bound EGFR, while we could not identify any ERBB2 peptides, suggesting that EGFR may be the primary partner of ERBB3 in these lung cancer cells with activating EGFR mutations. ERBB2 co-purified a large amount of protein groups, with EGFR being the most abundant followed by HS90A, HS90B, GRP78, and GRB2. ERBB2 may play a role in protecting EGFR against ubiquitination, and our results suggest strong interactions between EGFR and ERBB2 in these EGFR-mutated lung cancer cells (Shtiegman *et al*, 2007). Multiple members of the EGFR complex were co-purified with ERBB2, including EGFR, GRB2, HS90A and HS90B, CDC37, ERRFI, and UBS3B (Supplementary Figure S1D), resulting in our finding that ERBB2 could potentially play an important role in modulating EGFR signaling. CDC37 bound multiple kinases, including EGFR, A-Raf, CDK4, and IKK$\alpha_1$, suggesting chaperone requirements of these kinases in partnership with HSP90 and a possible functional importance in these cells.

## Identification of a core functional network in EGFR-addicted cells

We interrogated the 263-node interactome for proteins annotated to signal transduction pathways (ERB, MAPK, apoptosis, and cell cycle) using DAVID and identified 102 component proteins in this subset for further study (Figure 3B) (Huang da *et al*, 2009). Lung cancer cell lines were transfected with an siRNA library against 102 genes and subsequently assessed for changes in cell viability. We identified siRNAs that inhibited cell viability by > 50% (measured by CellTiter GLO assays) and had statistical significance (Supplementary Data File S3). To increase confidence in identified functional proteins and to minimize off-target effects, we interrogated three EGFR-addicted lung cancer cell lines (PC9, HCC827, and HCC4006). Pilot studies demonstrated that reproducibility across replicate biological experiments on 2 separate days was excellent (Spearman correlation coefficient = 0.968) (Supplementary Figures S2A and B). We were able to determine a 'core network' of proteins that significantly affected cell viability across the three EGFR-addicted cell lines (Figure 3C; Supplementary Figure S3). Fourteen proteins were shown to be important in maintaining cell growth across the three cell lines, including four original bait proteins (EGFR, GRB2, SHC1, and GRP78), one prey protein found in GRB2 and SHC1 pulldowns (ARHG5), three prey proteins of ERBB3 (PK3CA, P85B, and P55G), four protein kinases associated with CDC37 pulldowns (CD11A, CD11B, CDK9, and ARAF), one prey protein of GRP78 (GLU2B), and one protein identified in the phosphotyrosine analysis (MK12). When these proteins were mapped back onto the original EGFR interactome, we visually identified three functional clusters, in addition to EGFR, that strongly affected cellular proliferation in these EGFR-addicted lung cancer cells (Figure 3D, left panel), including proteins serving as components of the MAPK/Rho

pathway (SHC1, GRB2, MK12, and ARHG5), the PI3K pathway (PK3CA, P55G, and P85B), and kinases associated with CDC37 (CD11A, CD11B, CDK9, and ARAF). To examine the effects of target knockdown on downstream MAPK and PI3K activity, we performed in-cell western analysis of phosphorylated ERK and AKT, respectively (Figure 3D, right panel). Knockdown of EGFR, GRB2, SHC1, and MK12 had the most pronounced effects, suggesting important proximal roles of these proteins in orchestrating downstream signaling.

Knockdown of other proteins, such as ARHG5, appeared to affect other pathways.

CDC37, a co-chaperone with HSP90, was indirectly linked to survival given that many of the kinases associated with CDC37 were identified within the core network, including CDK9, CD11A, CD11B, and A-RAF. Furthermore, CDC37 is a known co-chaperone of activated kinases with HSP90 and HSP90 has been implicated in chaperoning aberrant RTK such as mutant EGFR and EML4-ALK in lung cancer (Shimamura et al, 2005;

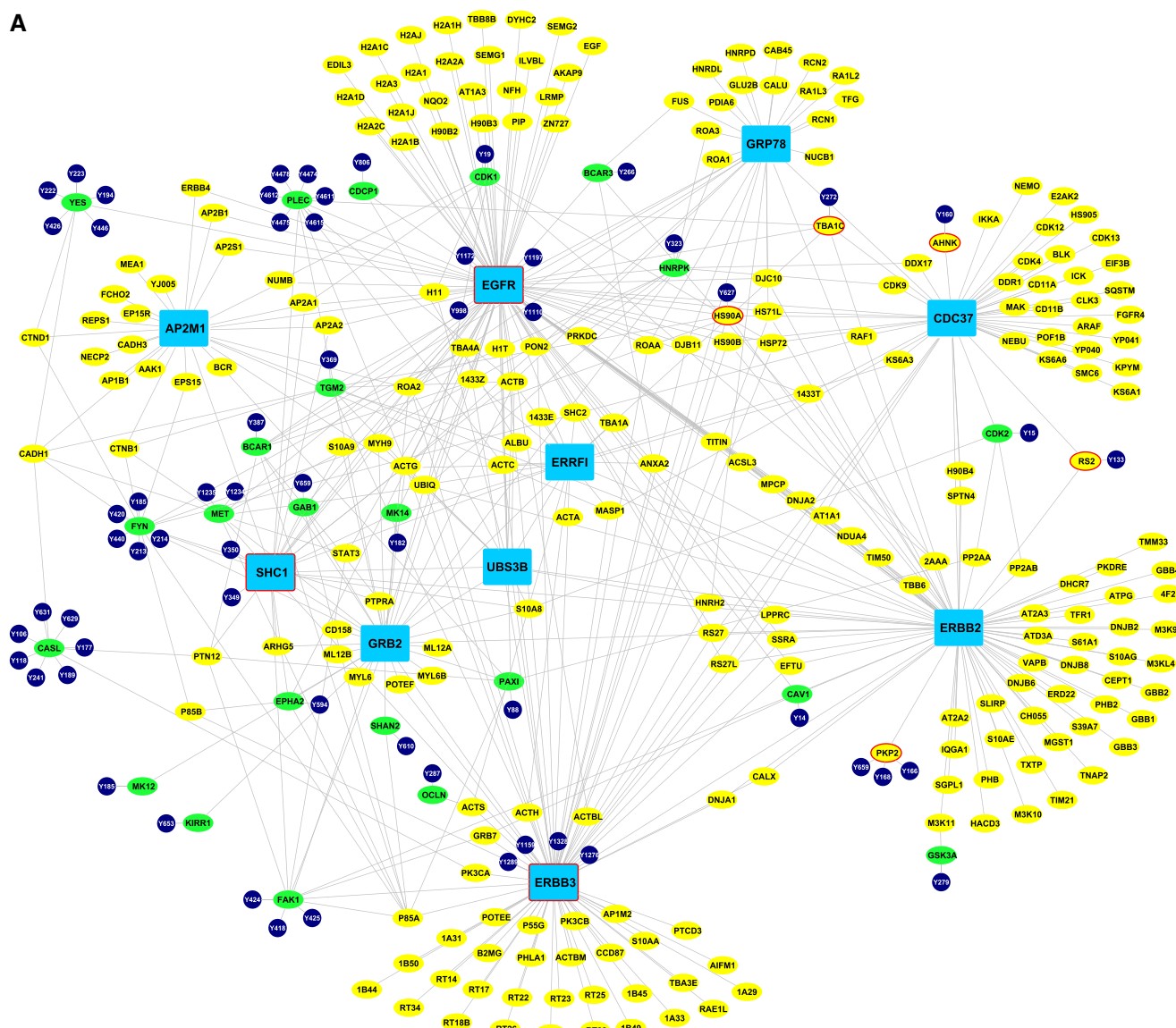

**Figure 3** Physical EGFR interactome generation and core network proteins identification. (**A**) The interactome ($N = 263$ proteins) was derived from TAP-LC-MS/MS experiments with bait proteins (blue rectangles). Yellow ellipse nodes indicate prey proteins directly identified from TAP experiments, while nodes with red border are from both pY and TAP experiments. Tyrosine phosphorylated proteins significantly perturbed by erlotinib identified from pY experiments, and are shown as green ellipses, including EGFR, SHC1, and ERBB3. Green ellipses indicate proteins identified from pY experiments that were added to the network based on the public interactions database. Phosphotyrosine sites significantly perturbed by erlotinib ($n = 62$) are indicated as blue circles connected to the relevant protein. Gray lines between proteins indicate bait–prey relationships derived by TAP experiments or literature-reported interactions between bait or prey proteins identified by TAP and pY containing proteins identified by phosphoproteomics. (**B**) Functional categories and number of proteins for each particular association. (**C**, top) Effects of RNAi analyses on cell viability across PC9, HCC827, and HCC4006 lung cancer cells and overlapping proteins represented in Venn diagram. (**C**, bottom) Characteristics of 14 significant proteins found in common among all 3 EGFR-mutant cell lines. (**D**) Core network proteins were mapped back onto the EGFR interactome. Right panels show effects of core network protein knockdown on ERK and AKT phosphorylation assayed through in-cell western analysis. Source data for this figure is available on the online supplementary information page.

**B**

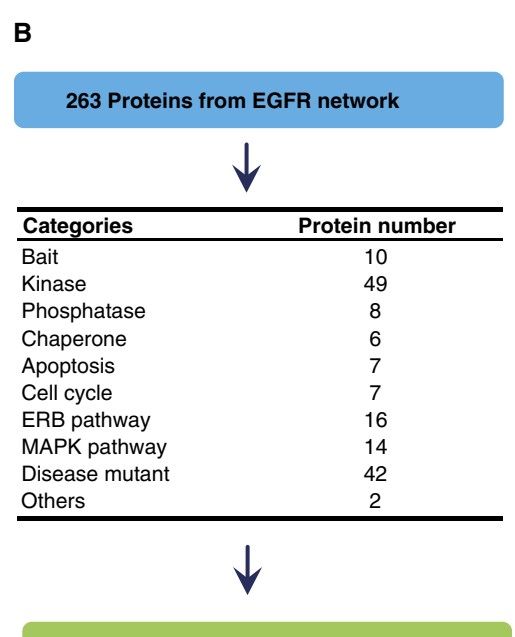

**263 Proteins from EGFR network**

| Categories | Protein number |
|---|---|
| Bait | 10 |
| Kinase | 49 |
| Phosphatase | 8 |
| Chaperone | 6 |
| Apoptosis | 7 |
| Cell cycle | 7 |
| ERB pathway | 16 |
| MAPK pathway | 14 |
| Disease mutant | 42 |
| Others | 2 |

**102 Candidates for custom RNAi library**

**C**

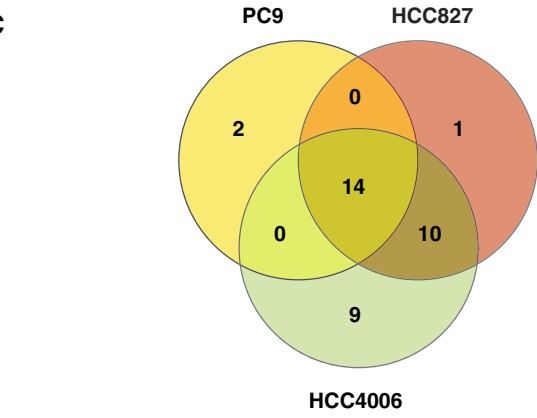

| Protein | Uniprot ID | Description | Source |
|---|---|---|---|
| EGFR | P00533 | Epidermal growth factor receptor | Primary bait |
| GRB2 | P62993 | Growth factor receptor-bound protein 2 | Secondary bait |
| SHC1 | P29353 | SHC-transforming protein 1 | Secondary bait |
| GRP78 | P11021 | 78 kDa glucose-regulated protein | Secondary bait |
| ARHG5 | Q12774 | Rho guanine nucleotide exchange factor 5 | Prey of GRB2 and SHC1 |
| PK3CA | P42336 | Phosphatidylinositol-4,5-bisphosphate 3-kinase 110 kDa catalytic subunit alpha | Prey of ERBB3 |
| P85B | O00459 | Phosphatidylinositol 3-kinase 85 kDa regulatory subunit beta | Prey of ERBB3 |
| P55G | Q92569 | Phosphatidylinositol 3-kinase 55 kDa regulatory subunit gamma | Prey of ERBB3 |
| CD11A | Q9UQ88 | Cyclin-dependent kinase 11A | Prey of CDC37 |
| CD11B | P21127 | Cyclin-dependent kinase 11B | Prey of CDC37 |
| CDK9 | P50750 | Cyclin-dependent kinase 9 | Prey of CDC37 |
| ARAF | P10398 | Serine/threonine-protein kinase A-Raf | Prey of CDC37 |
| GLU2B | P14314 | Glucosidase 2 subunit beta | Prey of GRP78 |
| MK12 | P53778 | Mitogen-activated protein kinase p38 gamma | Phosphotyrosine data |

**D**

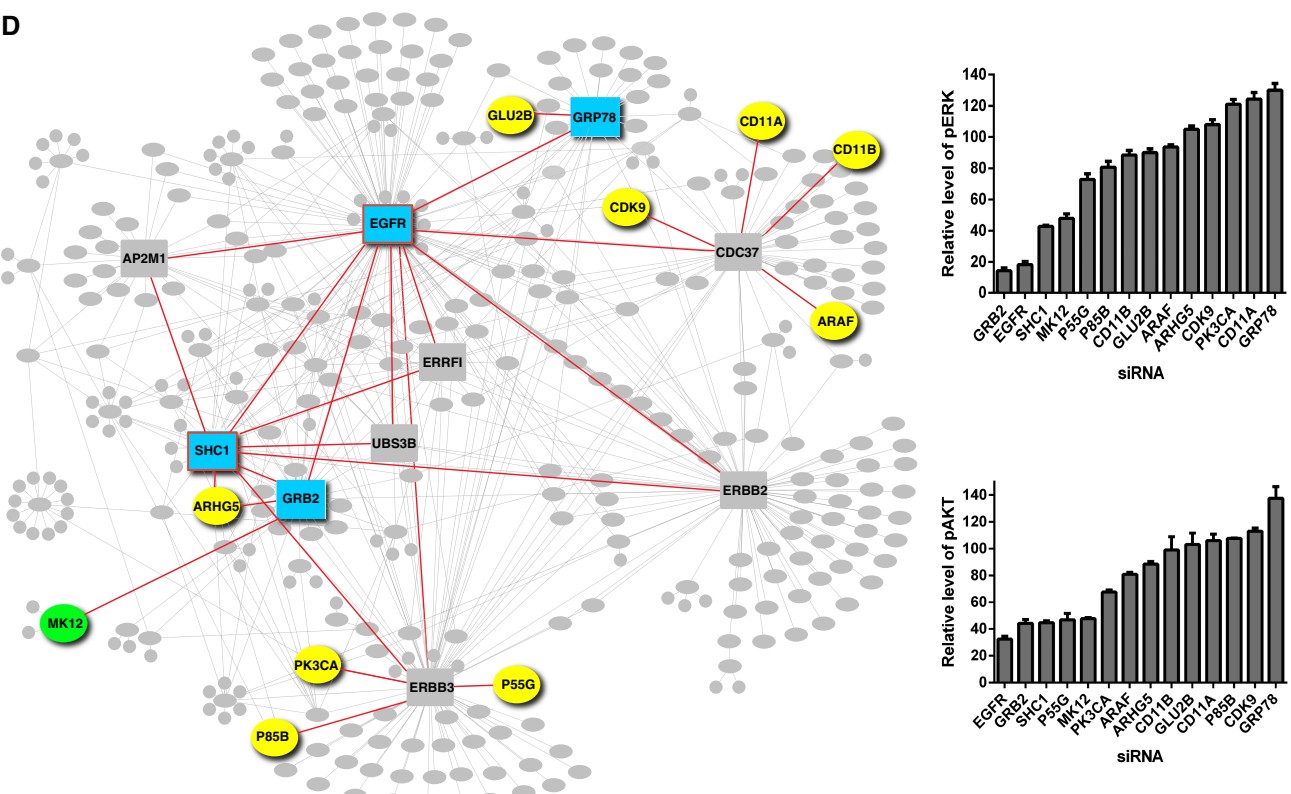

**Figure 3** Continued

Chen *et al*, 2010; Sequist *et al*, 2010a, b). CDC37 can also promote hyperproliferation when overexpressed (Stepanova *et al*, 2000). We examined effects of small-hairpin RNA (shRNA) knockdown in PC9 cells and in EGFR TKI-resistant PC9 cells harboring a T790M secondary mutation in EGFR (PC9GR). We identified two independent shRNAs that reduced levels of CDC37 following lentiviral infection (Figure 4A). CDC37 knockdown inhibited cell viability of both the

EGFR TKI-sensitive PC9 and the EGFR TKI-resistant PC9GR cells and induced cellular apoptosis (PARP cleavage) (Figures 4A and B). Knockdown of CDC37 reduced levels of EGFR and downstream pERK and pAKT, consistent with its role as a co-chaperone of EGFR with HSP90 (Figure 4C). The results with CDC37 are particularly intriguing as it suggests that chaperoned kinases in a tumor cell may enrich for a set of kinases necessary for cell survival. Similar results have been postulated with studies using HSP90 inhibitors in affinity capture studies and argue that approaches to capture client proteins can identify new oncoproteins and direct new targeted therapy approaches (Moulick *et al*, 2011).

We further tested the reliance of EGFR-mutated lung cancer cells on the core network by determining how known resistance mechanisms to EGFR TKI perturb the dependency of cells on the core network. We hypothesized that cells with secondary EGFR mutations mediating resistance that still retain EGFR dependence would similarly demonstrate continued dependence on the core network, while cells with other mechanisms would show lesser dependence on the core network. Parent PC9, HCC827, and HCC4006 cells were exposed to increasing concentrations of EGFR TKI, and surviving cells were identified and expanded. PC9GR cells

(gefitinib resistant) were shown to harbor a secondary T790M mutation in EGFR, whereas HCC827ER cells (erlotinib resistant) demonstrated MET overexpression, as previously reported (Engelman *et al*, 2007). HCC4006ER cells maintained EGFR mutation and became resistant to EGFR TKI through epithelial–mesenchymal transition (Suda *et al*, 2011). The EGFR TKI-resistant cells were transfected with pooled siRNAs of the entire library and examined for cell viability. We then compared the results from each pair to examine overlap of proteins affecting cell viability (Figure 5A). We confirmed equal and high levels of transfection efficiency across all six cell lines (Supplementary Figure S4).

As predicted, the PC9 and PC9GR cells showed high levels of overlap between the core network proteins, indicative of the common reliance of both cell lines on EGFR-driven interactome signaling. In HCC827ER cells with EGFR mutation and MET overexpression, the degree of overlap was reduced with loss of EGFR, GRB2, and SHC1 protein dependence. We interpreted our results to mean that EGFR signaling to MAPK through GRB2/SHC1 becomes less important in these cells that use MET to propagate survival signaling. This correlates with previous reports indicating that MET amplification is able to drive ERBB3-dependent activation of the PI3K pathway

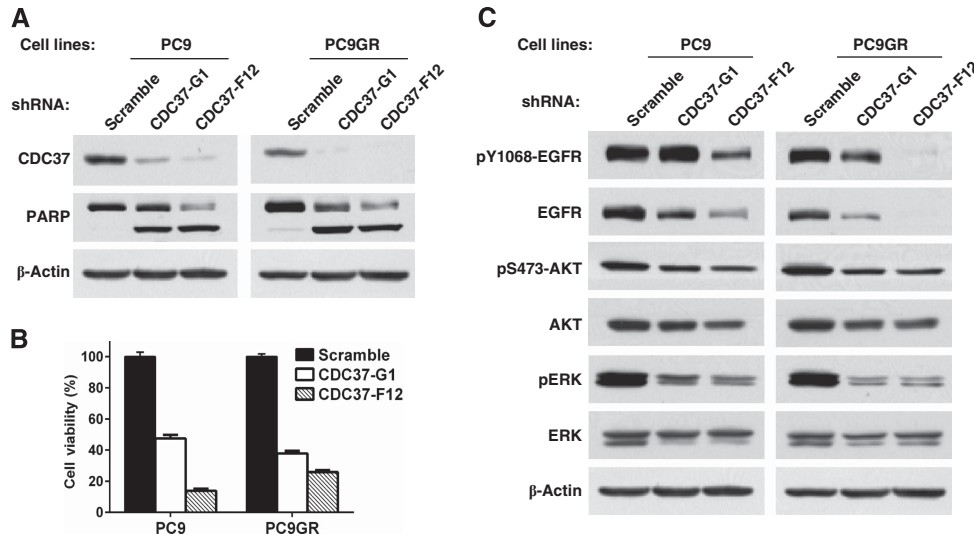

**Figure 4** CDC37 regulates survival in PC9 and erlotinib-resistant PC9GR cell lines through EGFR. (**A**) PC9 and PC9GR cells were infected with lentivirus expressing two distinct shRNAs against CDC37 or scrambled shRNA as a control. CDC37 depletion and PARP cleavage were evaluated by western blotting 72 h after infection. (**B**) Cell viability was assessed using CellTiter-Glo 5 days following infection. (**C**) Signaling changes were evaluated by western blotting 72 h after infection.

**Figure 5** Network protein vulnerability in EGFR TKI-resistant cells and specific for EGFR-mutated cells. (**A**) Core network protein perturbations track with EGFR TKI resistance mechanisms. For each pair of erlotinib-sensitive and -resistant cells, the protein name and number of significant effects from siRNA transfection in either parent cell line alone or overlapping with EGFR TKI-resistant cell line are shown in Venn diagram. The criteria for defined sensitivity to siRNA across all three pair cell lines required (i) the knockdown inhibited cell viability by >50% and (ii) had statistical significance ($P<0.05$) from three data points. (**B**) Lung cancer cell lines with EGFR-mutant ($n=5$) and wild-type EGFR ($n=12$) were transfected with pooled siRNA targeting 15 genes as described in Materials and methods. Cell viability was performed by CellTiter Glo 5 days after transfection. Heat map of the average values of cell survival from three technical replicates was plotted by the heatmap function in R (http://www.r-project.org/). The range and histogram of the average values are also shown at bottom right. The blue color indicates lower number of viable cells while red color indicates higher number of viable cells. *P*-values listed on the right side of the heat map were based on the Wilcoxon rank-sum test. The significant hits ('##': $P<0.01$; '#': $P<0.05$) are listed above the red line on the left side of the heat map. The cell lines labeled with '*' are EGFR mutant and resistant to EGFR TKI. (**C**) PC9 cells were exposed to 100 nM erlotinib for 1 h and DMSO as a negative control. Protein lysates were immunoprecipitated with anti-ARHG5 antibody and rabbit IgG as a control, then immunoblotted with indicated antibodies. (**D**) PC9 and PC9GR cells were infected with lentivirus expressing two distinct shRNAs against ARHG5 or lentivirus expressing non-targeting scrambled shRNA as a control. ARHG5 knockdown was detected by qPCR following 72-h infection (left panel). Cell viability assay was performed by CellTiter-Glo following 5-day infection (right panel). (**E**) PARP cleavage was evaluated by western blotting after 72-h infection and equal protein loading was confirmed by β-actin antibodies.

(Engelman *et al*, 2007). Finally, results with HCC4006ER cells were most dramatic, as HCC4006ER cells demonstrated independence to any of the core network proteins. Indeed,

published results indicate that this cell line is resistant to various signaling inhibitors, including inhibitors of MEK or PI3K (Suda *et al*, 2011). Conversely, we also searched

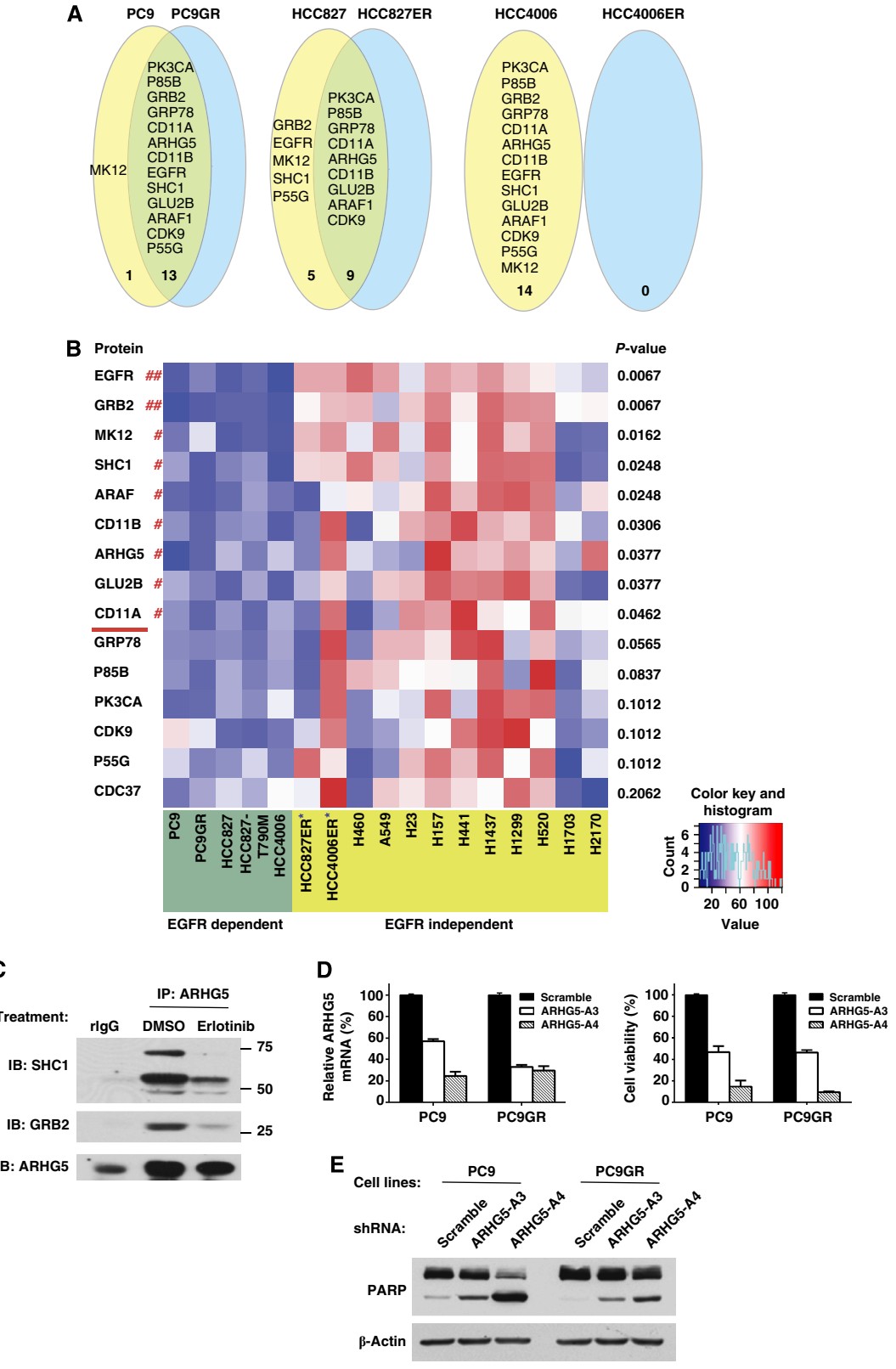

for siRNA that had more effects in the erlotinib-resistant cells compared to parent-sensitive lines (Supplementary Data File S3). We found six siRNA in PC9GR and three siRNA in HCC827ER having more effect compared to EGFR TKI-sensitive parental lines (PC9 and HCC827, respectively) while no siRNA was found to significantly affect HCC4006ER cells. OnlyAP2M1 and 1433Z were in common between the two cell lines (PC9GR and HCC827ER); however, the magnitude of effect was also rather modest. From these results, we conclude that functional interrogation of core network proteins acting as co-survival proteins with EGFR can give insight into resistance mechanisms to EGFR TKI. Furthermore, common proteins utilized by erlotinib sensitive and resistant (T790M EGFR) could be candidate targets to overcome resistance to EGFR TKI. For example, targeting ARHG5 would be able to inhibit growth of both erlotinib-sensitive PC9 cells and the erlotinib-resistant subclone (PC9GR), which is examined in more detail below.

To determine which of these 14 core network proteins are specifically involved in mutant EGFR-dependent survival as opposed to survival across wild-type EGFR lung cancer cell lines, we examined a larger battery of lung cancer cell lines for changes in viability following siRNA treatment. In addition to the six EGFR-mutant lines tested above (erlotinib-sensitive and -resistant pairs of PC9, HCC827, and HCC4006 cells), we added one additional engineered HCC827 cell expressing exon 19 deletion E746-A750 plus T790M, as well as 10 lung cancer cell lines lacking EGFR mutations (Soh *et al*, 2009). Cells were transfected with siRNA targeting the 14 core network proteins plus CDC37 and examined for cell viability. We confirmed equal and high levels of transfection efficiency across all of the cell lines (Supplementary Figure S4). The results of the siRNA analysis across the battery of EGFR-mutant and wild-type lung cancer cell lines ($N = 17$) are shown in Figure 5B and Supplementary Data File S3. Of the 15 siRNAs examined (14 core network plus CDC37), 9 siRNAs had significant differences in cell viability in the EGFR-mutated cells compared to wild-type EGFR cells. This included EGFR, GRB2, MK12, SHC1, ARAF, CD11B, ARHG5, GLU2B, and CD11A. These results suggest that the two chaperones included in our analysis (GRP78 and CDC37) have more generalized effects on cell survival. Similarly, components of the PI3K signaling cascade (P85B, PK3CA, and P55G) as well as the cyclin-dependent kinase CDK9 also have effects across multiple cell types independent of EGFR mutation. Interestingly, we also noted some similarity of the pattern of vulnerability of H1703 and H2170 cells to some of the core network proteins, including MK12, SHC1, and GLU2B. H1703 cells have been reported to be addicted to platelet-derived growth factor receptor-alpha (PDGFRα) through autocrine ligand secretion, whereas H2170 cells have ERBB2 amplification are sensitive to HER2 kinase inhibition (McDermott *et al*, 2007, 2009; Rikova *et al*, 2007).

ARHG5, which was identified as a novel core network protein affecting cell viability of all tested mutant EGFR cell lines, was found to be a protein demonstrating vulnerability in the EGFR-mutant lines compared to the wild-type lines. ARHG5 is a poorly studied protein in the family of Dbl-like guanine nucleotide exchange factors that in some systems and cellular contexts act upstream of the Rho family of small GTPases (Xie *et al*, 2005). Because of the novelty of this

protein, we performed further validation experiments to confirm the protein complexes by western blotting and used additional shRNA for additional functional studies. We identified ARHG5 from the lung cancer cell line TAP data as prey proteins of GRB2 and SHC1 baits. We confirmed that ARHG5 is associated with GRB2 and SHC1 in an EGFR kinase-dependent manner through immunoprecipitation of endogenous ARHG5 and western blotting against GRB2 and SHC1 (Figure 5C). We also validated our siRNA data by using lentiviral vectors expressing two independent shRNAs against ARHG5 and confirmed that both shRNAs reduced ARHG5 mRNA levels in PC9 and PC9GR cells (Figure 5D, left panel). Knockdown of ARHG5 was associated with reduced cell viability and with increased amounts of apoptosis based on PARP cleavage western blots (Figure 5E, middle and right panels). Finally, we examined whether ARHG5 could be directly affecting EGFR. We observed no consistent pattern with the two shRNAs for direct changes in either total EGFR or tyrosine phosphorylated EGFR (Supplementary Figure S5A). Similarly, we observed no changes in downstream ERK or AKT phosphorylation, agreeing with our previous studies using ARHG5 siRNA and in-cell western analysis of pERK and pAKT (Figure 3D, left panel). Studies using protein arrays for other downstream signaling proteins were unable to identify other mechanisms of action (Supplementary Figure S5B). These results, using both shRNA and siRNA studied in multiple EGFR-addicted lung cancer cell lines, suggest an important functional role of ARHG5 that requires further study as to the downstream pathways regulated by this protein.

## Chemical compounds targeting the core functional network

Lastly, we sought to determine whether we could identify therapeutic strategies, using existing compounds that target key proteins necessary for mutant EGFR survival. To identify compounds that can target key nodes related to survival, we exploited the conserved requirements of the core network to search across chemical space. We hypothesized that drugs impacting key nodes could be useful either alone or in combination with other inhibitors in EGFR TKI-resistant cell lines with T790M gatekeeper mutation (PC9GR, HCC827-T790M in Figure 5B). As shown above, these cells still retain a strong degree of dependency on the core network; thus, we reasoned that combination approaches could be generated. By including targets specific for EGFR-mutated lines (EGFR, GRB2, MK12, SHC1, ARAF, CD11B, ARHG5, GLU2B, and CD11A) as well as targets showing effects across EGFR-mutated and wild-type lines (GRP78,CDC37, P85B, PK3CA, P55G, and CDK9), we could test whether this strategy could likewise identify specific compounds versus those with generalized effects.

To bring together wide-ranging data sets of drug targets, we leveraged four resources for creating drug-target databases. We searched for the compounds and drugs that interact with core EGFR interactome proteins from four drug databases using (i) *in vitro* competition assays using purified kinase domains and kinase inhibitors (Davis *et al*, 2011), (ii) *in vitro* kinase assays (Anastassiadis *et al*, 2011), (iii) BindingDB

(http://www.bindingdb.org), and (iv) Drug Bank (http://www.drugbank.ca). BindingDB is a database that annotates ligand-protein complexes extracted from the literature focused on proteins that class drug targets (Liu et al, 2007; Wassermann and Bajorath, 2011). Drug Bank likewise is a database of drug and drug-target information (Knox et al, 2011). In total, we identified 1520 compounds that were reported to bind to or to inhibit one protein within the core network (Figure 6A). Not surprisingly, most of the compounds identified were directed against kinases, with 68.4% directed against EGFR and 21.8% against PK3CA (Supplementary Data File S4). ARHG5 and GLU2B, identified in our analysis as vulnerable proteins in EGFR-mutated cells, had no associated compounds, whereas still relatively few compounds existed that targeted SHC1 and GRB2. These results suggest an unmet need to develop chemical probes against these targets important in EGFR-driven lung cancer cells.

We next focused our attention on kinases within the core network, given the richness of the datasets as related to kinase inhibitors and the potential for near-term translation of results into patient-based trials using established kinase inhibitors. One of those drug databases (Davis et al, 2011) focused on the kinase inhibitors, which are either approved agents or under study in human clinic trials, and this database also provided data on drug binding affinity to targets (Kd value). We examined the 74 compounds reported by Davis with known interactions in kinases in our network and with stronger affinity to the targets (Kd < 100 nM). Because some databases report allele-specific interactions of compounds, we additionally searched for reported interactions between compounds binding EGFR proteins harboring the T790M gatekeeper allele. Although a number of compounds, including irreversible EGFR TKI, have activity against these proteins, the therapeutic window appears to be narrow, such that patients develop intolerable toxicity at levels of the compound necessary for kinase inhibition (Zhou et al, 2009; Sos et al, 2010). We therefore reasoned that combining EGFR TKI with the compounds specifically binding EGFR T790M gatekeeper allele or other compounds directed against the proteins within the core network could reveal synergies against cells harboring T790M and resistant to EGFR TKI.

We arrived at a drug network (Figure 6B) consisting of 26 compounds with reported interactions against 7 kinases within the core network. We identified 17 compounds affecting EGFR, including known irreversible EGFR TKI such as BIBW-2992 and CI-1033, as well as other compounds, such as dasatinib, that have effects on EGFR (Li et al, 2010). We also re-identified midostaurin (PKC412) and lestaurtinib (CEP-701), two compounds with FL3 activity and being evaluated with leukemia, as inhibitors with direct binding affinity to EGFR T790M alleles (Lee et al, 2013). On the basis of these findings, and the lack of other more prominent and specific compound:target interactions, we examined these potential T790M EGFR inhibitors in more detail. In vitro kinase assays confirmed that midostaurin has potent activity (IC$_{50}$ 1–2 nM) against EGFR proteins with T790M but has no substantial effect on EGFR kinases from wild-type alleles or activating point mutations (L858R) (Figure 6C). We confirmed in vivo interactions of midostaurin with EGFR T790M by examining differences in EGFR binding to midostaurin through drug

pulldowns. Protein targets of midostaurin were enriched using immobilized midostaurin and protein lysates from either PC9 or PC9GR cells. As predicted, EGFR bound to midostaurin in PC9GR cells with E746-A750 + T790M EGFR, but we found no EGFR associated with midostaurin in PC9 cells (Figure 6D; Supplementary Data File S5). Other validated midostaurin target proteins, such as AMPK1$\alpha$ and PDPK1, which are among the most prominent of the 71 protein kinases observed in total (Supplementary Data File S5), were recovered equally from cells with or without the T790M mutation (Davis et al, 2011).

On the basis of these results, we reasoned that combined erlotinib and midostaurin could overcome T790M-mediated resistance, as erlotinib would target wild-type and exon 19 deletion E746-A750 EGFR activity while midostaurin would inhibit exon 19 deletion E746-A750 + T790M EGFR. We used three lung cancer cell lines with T790M EGFR to test this hypothesis. This included PC9GR cells as well as two cell lines (HCC827 and HCC4006) that were engineered to have exon 19 deletion E746-A750 plus T790M EGFR through retroviral-mediated transduction. These results are shown in Figure 6E. Erlotinib or midostaurin alone had modest effects on total EGFR phosphorylation across the three cell lines; however, when combined, the two inhibitors had more pronounced effects on EGFR and downstream ERK phosphorylation. Combining erlotinib with lestaurtinib similarly reduced EGFR and ERK phosphorylation, while either agent alone had little to modest effects. Combining erlotinib and midostaurin or erlotinib and lestaurtinib significantly induced apoptosis compared with single drug treatment in cells with mutant EGFR but not those with wild-type EGFR (H157 cell line) (Figure 6F). Finally, evidence for synergism was observed combining erlotinib with either midostaurin or lestaurtinib in mutant EGFR lines with T790M EGFR, while no such effects were observed in two cell lines lacking EGFR mutation (Figures 6G and H).

## Discussion

We developed an integrated approach to experimentally identify interactomes driven by mutant EGFR proteins that can be exploited for therapeutic targets. We aimed to test the hypothesis that cancer-mutated EGFR would cause systems-wide effects by imposing severe modifications to the non-pathological network of molecular interactions. Analysis of this network would be useful to understand resistance mechanisms and potentially identify vulnerabilities that could be exploited therapeutically. Our studies comparing mutant EGFR proteins to wild-type EGFR proteins through TAP strongly support that these EGFR mutations associate the protein with MAPK activators and protect against trafficking and vesicular transport. Our results identified a core network consisting of 14 proteins required for the survival of multiple EGFR-addicted lung cancer cell lines. Importantly, we could identify conserved dependency in cells becoming resistant to reversible EGFR TKI through secondary gatekeeper mutation in EGFR (T790M), indicating the ability of these proteins to be targets in resistant tumors. We further defined specificity of components of the 14 core network for cells dependent on mutant EGFR for survival and identified EGFR, GRB2, MK12, SHC1, ARAF, CD11B, ARHG5, GLU2B, and CD11A as especially

vulnerable in EGFR-mutated and EGFR-dependent lung cancer cell lines. This list included the novel guanine exchange factor ARHG5 in addition to adaptor proteins GRB2 and SHC1. The targeting of oncogene-addicted cancers such as EGFR-driven lung cancer or ELM4-ALK-driven lung cancer has resulted in major improvements in patient outcomes, yet tumor regression can be short lived in some cases and acquired resistance is nearly always observed (Sequist *et al*, 2011). Temporal dynamics of pro-survival and pro-death pathways have been suggested to be important in oncogene addiction and response

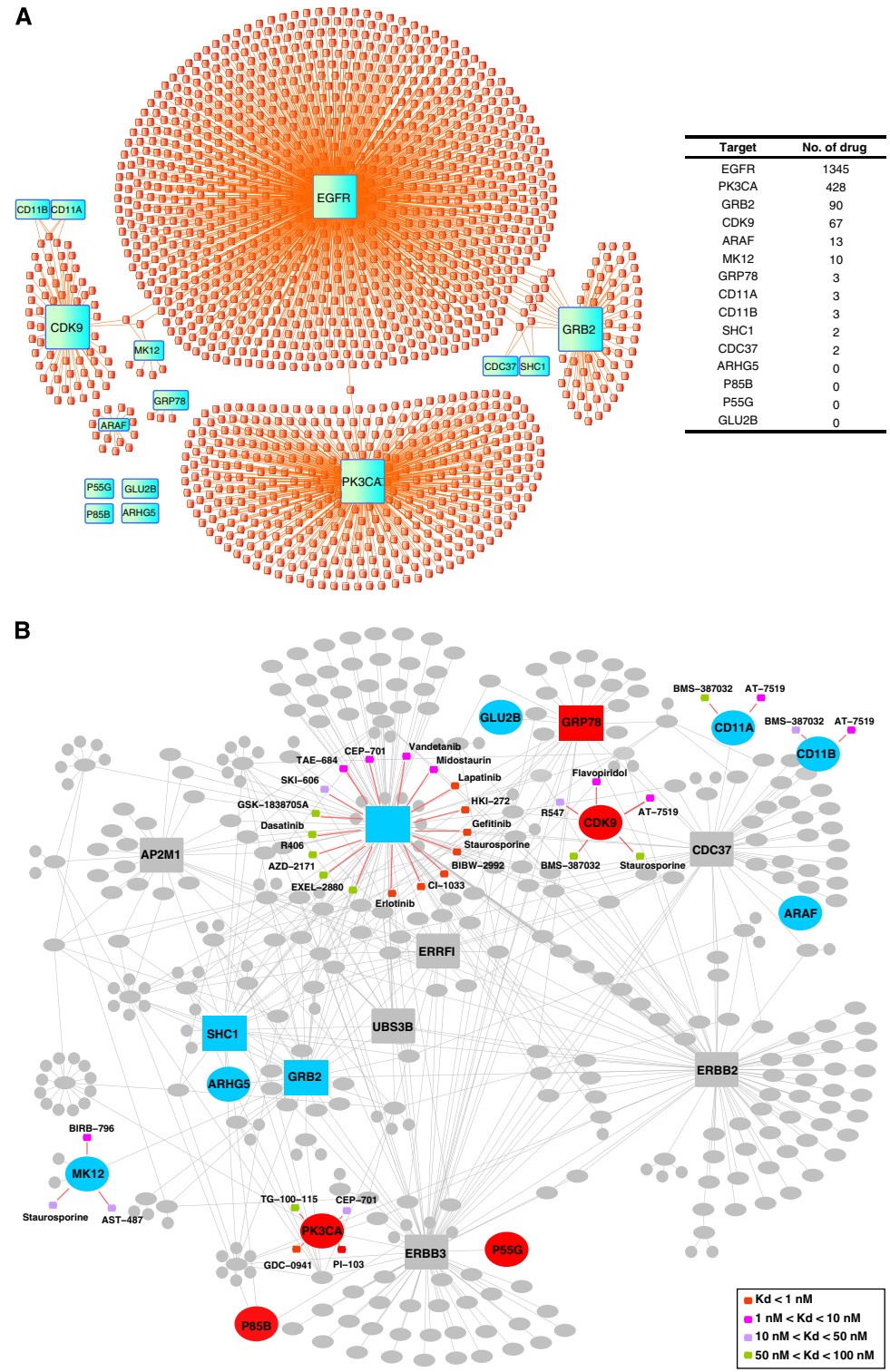

to targeted inhibitors (Sharma *et al*, 2006; Tran *et al*, 2011). Coupling of the components of the proximal signaling interactome identified by our approach to pro-survival and pro-death signaling could influence the process of oncogene addiction and could be important in predicting quantitative degrees of responses to kinase inhibitors. Components of this interactome should also be considered as potential drivers of acquired resistance in patients who have resistance mechanisms that have yet to be explained (Sequist *et al*, 2011).

The main studies consisted of the integrated proteomic studies coupled with functional analysis of proteins using siRNA and expanded analysis across additional cell lines. We hypothesized that interactome walking, starting with the mutant EGFR protein to define complexes, would enable a focused interrogation of proteins to define interacting proteins important in EGFR-mediated survival. We also evaluated whether adding additional experimentally derived phosphoproteomic data could provide additional targets. A total of 186

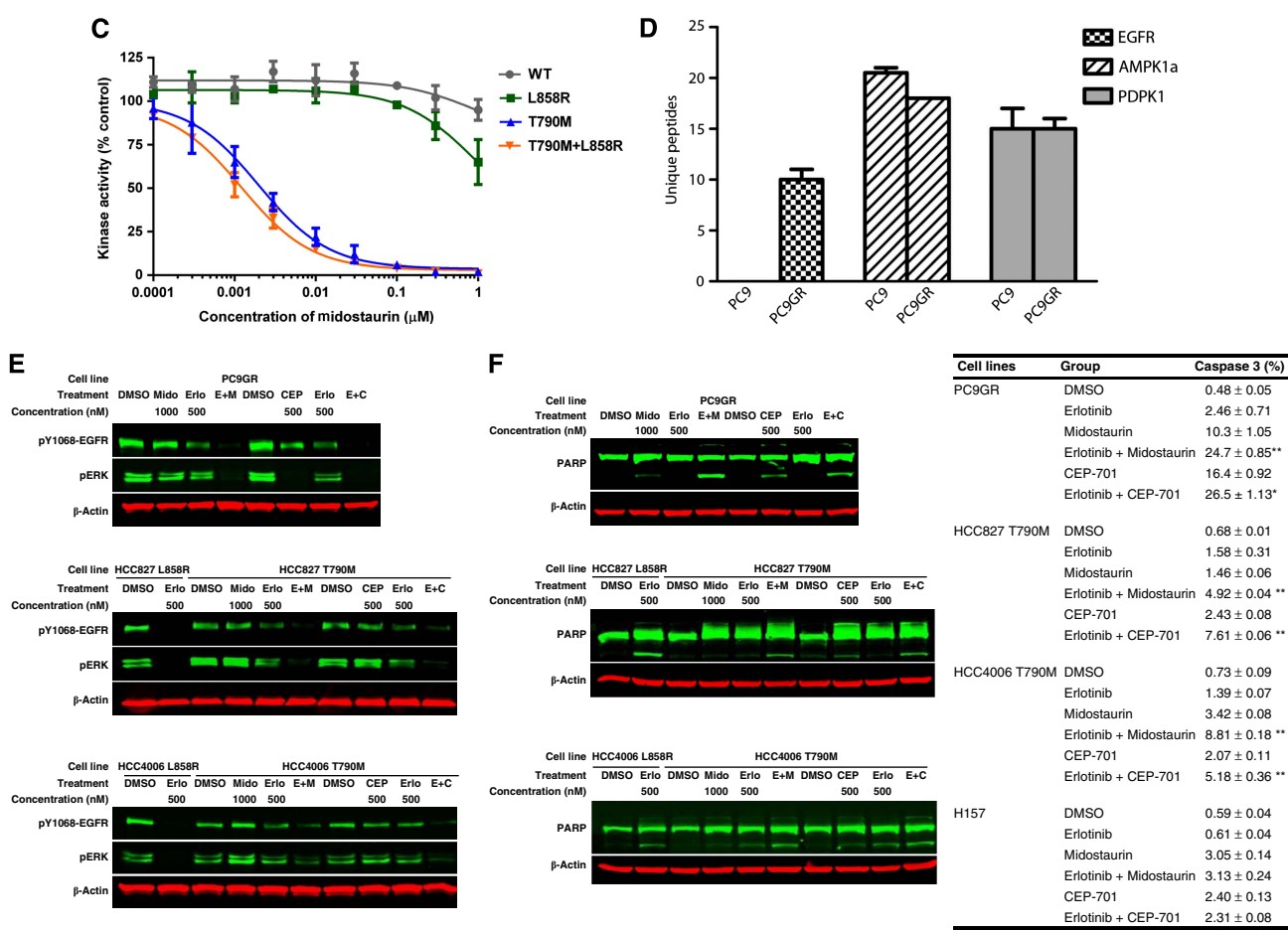

**Figure 6**    Continued.

**Figure 6**    Drug network derived from the core EGFR interactome and combination strategy to overcome the resistant cells. (**A**) Blue rectangles indicate core network proteins ($N = 14$), and 1520 red hexagon nodes represent compounds. Orange lines between drug and protein indicate the relationships derived by the literature-reported interactions. (**B**) Kinase inhibitors derived from competition binding assays were integrated to the core network. Blue nodes = EGFR-specific core proteins for cell survival and red nodes = non-specific core proteins in the EGFR interactome; rectangles = baits; ellipses = prey or protein identified by phosphoproteomics; orange lines = binding interactions between the inhibitor and the particular protein. The inhibitors are indicated by hexagons, with colors related to Kd shown at bottom right. (**C**) *In vitro* kinase assay results using purified EGFR kinases from indicated alleles were assessed after incubation with indicated concentration of midostaurin. Kinase activity results are shown as a percentage compared to DMSO control. (**D**) Lysates from either PC9 or PC9GR cells were incubated with immobilized midostaurin, and unique peptides corresponding to EGFR or other validated midostaurin target proteins (AMPK1α and PDPK1) were quantified. Number of unique peptides is shown for each protein. (**E**) The cells were exposed to erlotinib (Erlo), midostaurin (Mido), or combination at indicated concentrations. Signaling changes were evaluated by western blotting after 4 h treatment. (**F**) Indicated cells, including H157 cells with wild-type EGFR, were treated with 500 nM erlotinib, 1000 nM midostaurin, 500 nM CEP-701, or combination with erlotinib at the same concentration for 24 h. PARP cleavage was evaluated by western blotting (left panel) and caspase 3 activity measured by flow cytometry (right panel) are means ± s.d. for triplicate data points. '**': $P < 0.01$; '*': $P < 0.05$ comparing combination treatment with other single treatment. In (**E** and **G**), equal protein loading was confirmed by β-actin evaluation. (**G**) Cells were exposed to titration concentration of erlotinib, midostaurin, and combination of erlotinib with midostaurin or (**H**) Cells were exposed to titration concentration of erlotinib, CEP-701, and combination of erlotinib with CEP-701 at the same rate concentrations. The cell viability was assessed by CellTiter Glo after 5-day treatment. The growth curves were constructed by GraphPad Prism 6 and combination index (CI) was calculated by the CompuSyn software (http://www.combosyn.com/). CI < 1 represents synergism in drug combinations and was labeled below the curve of combination group.

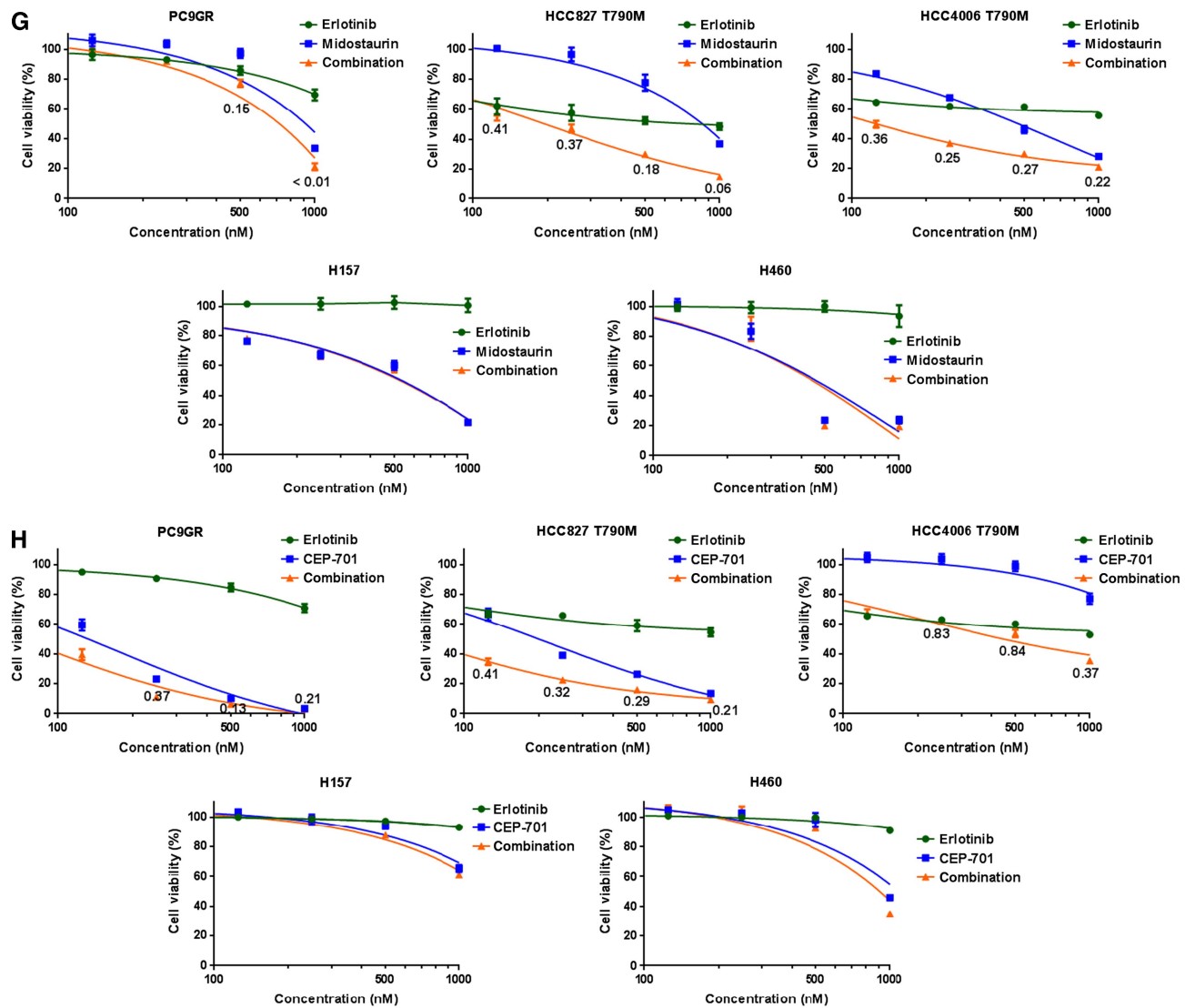

**Figure 6** Continued.

phosphotyrosine proteins were identified, of which 31 were added into the EGFR interactome *via* known protein–protein interactions, 19 were chosen for functional siRNA analysis, and 3 were identified as being critical for viability in EGFR-mutant cell lines (Supplementary Figure S6). At the end, we were curious to know whether the results could be directly translated into actionable therapeutic strategies and therefore linked to drug:protein databases. It was through this strategy and approach that we found midostaurin and lestaurtinib as reported T790M EGFR inhibitors. This was done through the described database searching. While it is true that the databases reflect the literature, it can be very difficult to identify all relevant papers in the literature and mine data sets that are frequently found in Supplementary material. This, we would argue, is a strength of our approach, as we found these important interactions driven by the strategy of linking our analysis results to the database of drug targets. Nonetheless, we acknowledge that, while inspired by the interactome analysis, one could have accomplished the same

discovery of the drug synergy by focusing initially on database searches of T790M EGFR interacting compounds. We were surprised that midostaurin had allele-specific EGFR targeting properties, since this drug has been promoted mainly as an FLT3 and PKC inhibitor. This prompted the chemical proteomics validation data and the additional validation data. Our results with midostaurin suggest the possibility of dual attacks on EGFR, as demonstrated in previous studies combining small-molecule TKI and monoclonal antibodies against EGFR (Regales *et al*, 2009). The mean trough concentrations of midostaurin have been reported to be ∼1.25–1.5 mM, which in our cell models can inhibit EGFR phosphorylation and downstream ERK phosphorylation (Fischer *et al*, 2010). Despite its promiscuous nature, the drug has been reported to be well tolerated in patients with leukemia and lung cancer (Monnerat *et al*, 2004; Fischer *et al*, 2010). Similarly, lestaurtinib, another staurosporine derivative, has mainly been studied in leukemia where it has demonstrated good tolerability with mean trough concentrations of ∼3.75 mM

achievable with twice daily dosing that again exceeds the concentration necessary for EGFR phosphorylation inhibition in cells (Marshall *et al*, 2005). Our results suggest that multiple targets of lestaurtinib could be active and important in the lung cancer cells studied, as we observed reductions in ERK phosphorylation at concentrations below those necessary for inhibition of EGFR phosphorylation. Recently, another study has suggested that ability of midostaurin to be an EGFR T790M inhibitor (Lee *et al*, 2013). However, our results suggest that optimal cell killing will occur through combined effects of midostaurin and another EGFR inhibitor that retains activity against non-T790M EGFR.

It is important to highlight the successes and limits of this integrated approach starting with interactome characterization by mass spectrometry followed by RNAi analysis and linkage to drug-target networks. We show the ability to identify and therefore nominate targets that are necessary for mutant EGFR-mediated survival signaling. This in turn lead to interrogation of drug databases that collated large-scale drug-target data sets that can be difficult to mine and uncovered two compounds with activity against T790M EGFR isoforms. However, the approach does have limits, which we acknowledge, which could be improved on in future work. First, a number of nominated targets are not typical druggable targets with enzyme activity but rather consist of adaptor proteins lacking enzyme activity. Most of the key proteins had no reported compounds or had compounds not useful in clinical studies. For example, no reported compounds were identified that interfere with ARGH5, identified in our analysis as being especially vulnerable in EGFR-mutated cells. Nonetheless, recent re-emerged attempts to drug the undruggable, including interfering with protein–protein interactions or covalently modifying proteins facilitating their degradation, should reprioritize these targets for future drug discovery efforts, as successful compounds will have molecular diseases to attack (Neklesa *et al*, 2011; Patgiri *et al*, 2011; Neklesa and Crews, 2012). Second, some of the nominated targets found in the functional analysis of the interactome have generalized effects across EGFR-dependent and -independent lines. This could be positive, however, as we can also therefore deprioritize targets that may lack therapeutic windows or have non-specific effects across multiple tumor types. Finally, lack of specificity of drugs or promiscuous that extends to non-specific targets also limits the approach. For example, compounds were identified that attacked specific proteins displaying vulnerability for EGFR-mutant cells (CD11A/B) yet also have effects on proteins (CDK9) that have generalized anti-tumor effects across lung cancer cell lines. Better results can be anticipated with increased high-quality drug:target data sets, as we used databases from the public domain that may lack more novel compounds or lack better specific compounds. Further examination of other drugs and targets identified through our analysis may reveal additional co-targeting strategies.

In summary, we demonstrated the validity of interactome walking to identify key nodes within a survival network driven by a mutant oncogene that when linked to well-annotated drug-target databases can identify new therapeutic strategies. We were able to nominate proteins that are specifically involved in mutant EGFR survival, while conversely able to identify target proteins necessary for growth and survival of many cell types. Further studies are necessary to maximize the clinical utility of this approach, either by developing new drugs targeting key, as yet 'undruggable' targets, or by exploiting chemical promiscuity to devise rationale combination therapy.

## Materials and methods

Details on Materials and methods are found in Supplementary Materials.

### Cell lines

Sources of lung cancer cell lines are as previously described (Li *et al*, 2010). All cell lines have been maintained in a central bank at Moffitt since 2008. All cell lines have been authenticated by STR analysis (ACTG Inc, Wheeling, IL) as of September 2010, and all are routinely tested and negative for mycoplasma (PlasmoTest, InvivoGen, San Diego, CA).

### TAP, phosphoproteomics, and data analysis

TAP was performed as described (Haura *et al*, 2011) and phosphopeptide immunoprecipitation and purification were performed, with results analyzed by nano-LC-MS/MS as described previously (Li *et al*, 2010). The protein interactions from this publication have been submitted to the IMEx (http://www.imexconsortium.org) consortium through IntAct (PMID: 22121220) and assigned the identifier IM-21413. Phosphoproteomics data sets have been uploaded to PeptideAtlas (http://www.peptideatlas.org) with identifier PAS00319. Detailed methods can be found in Supplementary Materials and Methods. Peak selection and conversion of RAW files into MGF format for subsequent protein identification were done by a combination of XCalibur (Thermo Scientific, Waltham, MA) and Trans Proteomics Pipeline (Keller *et al*, 2005) software tools. For the initial protein search, our group used Mascot (version 2.3.02, www.matrixscience.com) with 10 p.p.m. parent and 0.6-Da fragment mass tolerance. Searches were limited to fully tryptic peptides with maximum of 1 missed cleavage, carbamidomethyl cysteine as fixed modification, and methionine oxidation as variable. Mascot peptide score threshold was set to 30, and at least three peptide identifications per protein were required. Searches were run against the human component of UniProtKB/SwissProt database (version 57), including all protein isoforms. The initial peptide identifications were used to deduce linear transformations for parent and fragment masses that would minimize the mean square deviation of measured masses from the theoretical ones. Calibrated peak files for TAP and phospho-enriched samples were searched against the same human proteins database by a combination of Mascot and Phenyx (version 2.5.14 by GeneBio, Switzerland) search engines. Phosphorylations of serine, threonine, and tyrosine were included into the list of variable modifications for the analysis of phospho-enriched samples. The results of the two search engines were merged, requiring at least two distinct peptides with a score above threshold of either search engine. We also accepted single peptide hits, if the score was above a more stringent threshold (see Supplementary Table S1). Spectra with conflicting peptide identifications were excluded from the combined result. Scores and mass tolerances were tuned to achieve 1% FDR of peptide identifications when searched against the reversed sequences database (see Supplementary Table S1). The other parameters were the same as for the initial search. For grouping of proteins based on shared peptides, identifications from all replicate MS runs were pooled together, and the proteins without protein-specific peptides were discarded. For data set comparison and bait–prey interaction network construction, isoform information of the identified proteins was discarded. Pulldowns with tagged GFP were used as negative controls for other TAP pulldowns. Proteins identified in these pulldowns were considered as non-specific binders and removed from the resulting network. Each identified phosphorylation site was assigned a $\chi^2$-based *P*-value for the hypothesis that erlotinib treatment does affect the number of spectra for peptides that contain a given site in phosphorylated or

unphosphorylated states:

$$p = P\left(x \ge \chi^2 \begin{pmatrix} N_{e+,Ph+} & N_{e+,Ph-} \\ N_{e-,Ph+} & N_{e-,Ph-} \end{pmatrix} \right),$$

where $N_{e+,Ph+}$ is the number of spectra for peptides containing a phosphorylated site in a sample treated with erlotinib. Significance threshold was set to $P = 0.1$. The significant phosphorylation sites of proteins that were also detected in TAP pulldowns were mapped directly to the protein–protein interaction network. For the other sites, the interactions of phosphorylated proteins with TAP proteins were searched using an internal database that aggregates public protein–protein interaction resources (IntAct, BioGrid, MINT, HPRD, and InnateDB). Phosphorylated proteins that were found to interact (be physically associated) with TAP proteins were added to the network together with their significant sites of phosphorylation.

## Immunoprecipitation and protein expression analysis

Western blotting was performed as previously described (Li *et al*, 2010). Primary antibodies used in these studies consisted of EGFR, pTyr1068-EGFR, PARP, p44/42 MAPK, pThr202/Tyr204-p44/42 MAPK, AKT, pSer473-AKT, and tSRC from Cell Signaling (Beverly, MA); GRB2, STAT3, and ERRB3 from Santa Cruz Biotechnology (Santa Cruz, CA); HA, ERRFI, and β-actin from Sigma (St Louis, MO); SHC1 and CDC37 from Thermo Scientific (Rockford, IL); ERBB2 from Millipore (Billerica, MA); UBS3B (STS-1) from Rockland (Gilbertsville, PA); and IRDye™ 800CW-labeled goat-anti-rabbit and IRDye™ 680-labeled goat-anti-mouse secondary antibodies from LI-COR (Lincoln, NE).

## Cellular assays

Cell viability (CellTiter-Glo) assay was conducted according to the manufacturer's recommendations for the CellTiter-Glo Luminescent Cell Viability Assay (Promega, Madison, WI). Apoptosis assays were performed according to the manufacturer's recommendations of PE-conjugated Monoclonal Active Caspase-3 Antibody Apoptosis Kit from BD Pharmingen (San Diego, CA). In-cell western blotting was performed using IRDye™ 800CW-labeled goat-anti-rabbit along with IRDye™ 680-labeled goat-anti-mouse secondary and imaged using the LI-COR Odyssey Infrared Imaging Scanner in both 700- and 800-nm channels.

## RNA interference

The ON-TARGET plus Smart pool custom siRNA library included siRNA pools containing four different siRNAs. This custom library and the ON-TARGET plus Non-Targeting Pool, non-targeting siRNA #4, and GAPD control siRNA pool were purchased from Thermo Scientific (Dharmacon). siRNA was delivered in triplicate to each well using a Precision™ microplate liquid handler (BioTek). Plates were incubated for 5 days for cell viability assay and for 48 h for the pERK and pAKT in-cell western analyses. Data analysis was automated with RNAither (Rieber *et al*, 2009). Viability and signaling changes were determined for each target gene after normalization on ON-TARGET plus Non-Targeting pool control siRNA. For viability, significant hits were defined as (1) inhibition of cell viability $> 50\%$ and (2) $P$-value $< 0.05$ using Student's *t*-test.

## Drug networks

Compounds that interact with core EGFR interactome proteins were identified from four drug databases using (i) *in vitro* competition assays using purified kinase domains and inhibitors (Davis *et al*, 2011), (ii) *in vitro* kinase assays (Anastassiadis *et al*, 2011), (iii) BindingDB (http://www.bindingdb.org), and (iv) Drug Bank database (http://www.drugbank.ca) We filtered search results by applying cutoffs for Kd ($< 100$ nM for databases (i) and (iii)) or Ki, Kd, IC$_{50}$, or EC$_{50} < 10$ nM for BindingDB results. To appropriately take advantages of the

Anastassiadis screening data, a network-based method was derived to produce more robust ranking scores. An 'EGFR-specific' network based on TAP experiments, in which the 'EGFR-specific' interactions among the profiled kinases as potential drug targets were obtained. On the basis of the drug screening values from Anastassiadis *et al*, all potential drug candidates are connected to different kinases in this EGFR-specific network. The final ranking scores were derived by diffusing the given screening values in the Anastassiadis data based on the short distances between potential drug candidates to different kinases. The underlying assumption is that the drug candidates can be more effective for a specific kinase if they have strong effects on its close neighbors in this 'EGFR-specific' network. On the basis of these derived ranking scores, a cutoff threshold value was selected to derive the final lists of candidate kinase targets and the corresponding drug candidates. Drug–protein interaction networks were generated by the Cytoscape software, converted, loaded into yEd Graph Editor (yWorks, Tübingen, Germany), and visualized based on the Fruchterman-Reingold layouting algorithm, which provides the proportion of the drugs that target a particular protein.

## Chemical proteomics

c-Midostaurin was synthesized by derivitization of staurosporine with $N$-Boc-protected $m$-aminomethylbenzoic acid in the presence of HATU and diisopropylethylamine and subsequent deprotection with trifluor-oacetic acid (all from Sigma-Aldrich). c-Midostaurin was immobilized on NHS-activated Sepharose 4 Fast Flow resin (GE Healthcare Bio-Sciences AB, Uppsala, Sweden) as described (Rix *et al*, 2007). Affinity chromatography was performed as reported previously (Rix *et al*, 2007). LC-MS/MS analysis was performed in duplicate on a LTQ-Orbitrap MS (Thermo Fisher). The human UniProt database was appended by peptides corresponding to EGFR mutations known to be present in PC9 and PC9GR cells, and an EGFR inclusion list was defined before searching was conducted with the Mascot (Matrix Science) and Sequest (Thermo Fisher) search engines.

## Kinase inhibition analysis

*In vitro* kinase inhibition assays were performed on the Millipore KinaseProfiler™ platform. Midostaurin was compared to DMSO for inhibition of the different alleles of EGFR, including wild type, L858R, T790M, and T790M plus L858R at 0.0001, 0.0003, 0.001, 0.003, 0.01, 0.03, 0.1, 0.3, and 1 μM.

## Supplementary information

## Acknowledgements

We thank Matthias Gstaiger (Zurich) for providing strep-HA TAP tags, members of the Center for Molecular Medicine (CeMM) and the Superti Furga laboratory for helpful discussions, Fumi Kinose for help with cell culture, and Rasa Hamilton (Moffitt Cancer Center) for editorial assistance with the manuscript. The work was partially funded by grants from the National Functional Genomics Center (W81XWH-08-2-0101) and the NIH (Moffitt Lung Cancer SPORE P50-CA119997). This work has been supported in part by the Proteomics, Molecular Genomics, Analytic Microscopy, and Flow Cytometry Core Facilities at the H. Lee Moffitt Cancer Center & Research Institute (Tampa, FL). AS and JC are supported by a Bioinformatics Network (BIN III) grant of the GEN-AU program of the Austrian Ministry for Science and Research.

*Author contributions:* JL originated the study questions, performed experiments, analyzed data, wrote the manuscript and approved the final draft; KB analyzed mass spectrometry data related to TAP experiments, analyzed data, and approved the final draft; AS analyzed mass spectrometry data, wrote the manuscript, and approved the final

draft; BF analyzed mass spectrometry data related to phosphoproteomics and approved the final draft; TY performed experiments and approved the final draft; GZ performed experiments and approved the final draft; IO provided drug-resistant cell lines and approved the final draft; J-YK performed experiments and approved the final draft; LS performed experiments, analyzed data, and approved the final draft; BR analyzed RNAi data; YB performed experiments and approved the final draft; MS analyzed RNAi data and approved the final draft; FG provided technical expertise related to TAP and TAP vectors and approved the final draft; XQ analyzed interactome data and approved the final draft; GW developed coupled midostaurin; UR performed drug pulldown experiments, analyzed the data, wrote the manuscript, and approved the final draft; SE analyzed data and approved the final draft; JC analyzed the data and approved the final draft; JK analyzed the data and approved the final draft; GSF originated the study questions, wrote the manuscript, and approved the final draft; EBH originated the study questions, performed TAP experiments, analyzed the data, wrote the manuscript, and approved the final draft.

## Conflict of interest

The authors declare that they have no conflict of interest.

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
