## [Review Process File · Molecular Systems Biology]

Perturbation of the mutated EGFR interactome identifies vulnerabilities and resistance mechanisms

Jiannong Li, Keiryn Bennett, Alexey Stukalov, Bin Fang, Guolin Zhang, Takeshi Yoshida, Isamu Okamoto, Jae-Young Kim, Lanxi Song, Yun Bai, Xiaoning Qian, Bhupendra Rawa, Michael Schell, Florian Grebien, Georg Winter, Uwe Rix, Steven Eschrich, Jacques Colinge, John Koomen, Giulio Superti-Furga, and Eric B. Haura

Corresponding author: Eric B. Haura, H. Lee Moffitt Cancer Center and Research Institute

Review timeline:

Submission date:	19 February 2013
Editorial Decision:	28 March 2013
Revision received:	26 July 2013
Editorial Decision:	09 September 2013
Revision received:	26 September 2013
Accepted:	02 October 2013

Editor: Maria Polychronidou

Transaction Report:

1st Editorial Decision

28 March 2013

Thank you again for submitting your work to Molecular Systems Biology. We have now heard back from the three referees who agreed to evaluate your manuscript. As you will see from the reports below, the referees find the topic of your study of potential interest. They raise, however, substantial concerns on your work, which should be convincingly addressed in a revision of the manuscript.

Overall, the reviewers appreciate your highly integrative analysis and the systematic collection of a vast amount of data. Many of the reviewers' comments refer to the need to clarify and better document several points throughout the manuscript. However, the reviewers also raised several major points, which should be carefully addressed. Among the more fundamental issues are the following:

- Additional experimentation is required in order to address whether the effect of the knockdowns and drug treatments is specific for an EGFR mutant context.
- The integration of the phosphotyrosine proteomics data needs to be described in detail and its contribution to the findings should be clearly shown.
- The validity of the combinatorial therapy approach should be demonstrated beyond the single cell line used.

If you feel you can satisfactorily deal with these points and those listed by the referees, you may wish to submit a revised version of your manuscript. Please attach a covering letter giving details of the way in which you have handled each of the points raised by the referees. A revised manuscript will be once again subject to review and you probably understand that we can give you no guarantee

at this stage that the eventual outcome will be favorable.

REFeree REPORTS

Reviewer #1:

The authors constructed a comprehensive interaction network of EGFR mutant and used it to explore drug resistance mechanisms and therapeutic opportunity. To this end, they performed tandem affinity purification-liquid chromatograph-mass spectrometry (TAP-LC-MS/MS). They then determined EGFR core networks that catalog proteins that highly affected the cell viability of cancer by using RNA interference screening. By profiling the perturbation effects of the core network and comparing them between drug-resistant cancer cell lines, they showed a strategy to elucidate drug resistance mechanisms. Finally they suggested a few drug combinations for potential cancer therapy.

I have some points that the authors should address.

- 1) In the abstract and page 11, the authors mentioned three modules (functional clusters) referring Figure 3C. However it is not recognizable from Figure 3C. The authors should clearly state how they recognized three distinct functional clusters.
- 2) On page 8, the authors chose GRB2, SHC1, ERFFI, UBS3B, ERBB2, GRP78, AP2M1, and CDC37 for the second round of TAP based on their roles in linking EGFR to downstream MAPK signaling. This selection strategy can cause bias to the EGFR interactome networks and subsequent analysis. All the proteins from the first round of TAP should be used as bait in the second round of TAP. At least they should be chosen in an unbiased way.
- 3) On page 9, it should be described more clearly how the phosphotyrosine proteomics data were integrated. Even in the supplementary method section, it was not described in enough detail. It should be documented whether tyrosine kinase and SH2 domains were considered and how their specificities were considered when the authors added interactions.
- 4) The authors performed tyrosine phosphoproteomics only in PC9 cells. They should justify EGFR-driven phosphorylation signaling is not different between PC9 and the other cells the authors used, such as HCC827 cells. Otherwise, they also should perform phosphoproteomics in other cells.
- 5) On pages 11 and 12, the authors should provide a more reasonable justification of why they picked up CDC37 and ARHG5. For CDC37, they should provide logical ground or evidence when they say "Although cell viability for CDC37 was only 54%, its association with a module of protein kinases driving cell viability identified in the core network suggested an important functional role."
- 6) On page 10, the authors used DAVID to identify proteins annotated to signal transduction pathways (ERB, MAPK, apoptosis, and cell cycle). The selected genes were subjected to the siRNA screening to determine the "core network". Using gene annotations to select candidate genes is highly risky since it causes biases to the subsequent analysis and prevents novel findings. The authors should use all the 265 genes for the siRNA screening to avoid potential bias or at least select candidate genes in an unbiased way.
- 7) On the sentence in page 12 "Knockdown of CDC37 reduced levels of EGFR and downstream pERK and pAKT, consistent with its role as a co-chaperone of EGFR with HSP90", did the author mean pEGFR instead of EGFR?
- 8) On page 15, the authors should state how they integrated in vitro kinase assays (Anastassiadis et al, 2011), since it does not provide kD values. Also these types of data are highly erroneous, a benchmark needs to be provided.
- 9) On page 17, the authors should justify why they chose AT-7519, BMS387032, flavopiridol, and BKM120.

10) Throughout the manuscript, there are a number of unclear sentences, which need to be revised or explained in more detail.

a. On page 12, "ARHG5, which was identified as a novel core network protein affecting sensitivity of all tested mutant EGFR cell lines"

b. On page 14, "common proteins used by EGFR (including T790M EGFR) in resistant cells may be considered targets to pursue in overcoming resistance to EGFR TKI, such as ARHG5 and CDC37, as we have shown above"

c. On page 17, "we examined efficacy of compounds reported to affect survival kinases located within the core network in EGFR TKI-resistant cells harboring T790M"

Minor comments

1) Mixed use of 'Figure #' and 'Fig. #' throughout the manuscript.

2) When the authors say "the degree of overlap was reduced with loss of EGFR, GRB2, and SHC1 protein dependence" (page 14), it seems they omitted P55G.

3) On page 17, kinase-inhibitor interactions were stated in a confusing way.

4) On Figure 3C, the meaning of the gray boxes are unclear.

5) On Figure 4, it would be helpful to state the meaning of 'SC', 'G1', and 'F12'.

6) On Figure 5, it would be more comprehensible if the two 'ER's are replaced by full names ('HC827ER' and 'HCC4006ER').

Reviewer #2 :

Summary

Li et al. conduct an integrative analysis of the EGFR protein interaction network. They identify a core network of 14 proteins, which is critical to the survival of several EGFR-mutated lung cancer cells. Interestingly, cells with acquired resistance to EGFR inhibition demonstrated differential dependence on the core components. Integrating their interactome data with a drug target network the authors identify and validate (co-)targeting strategies for acquired EGFR inhibitor resistant cells.

Pros

-The authors present a highly integrative analysis strategy combining interactome mapping, phosphoproteomics, and siRNA screens with targeted follow-up experiments.

-Combination therapy with drugs targeting the core network dramatically reduced cell viability of EGFR TKI resistant cells.

Cons

-The work lacks some important controls (e.g. in non-mutant EGFR lung cancer cells, and in normal-like cells) to test for general network component knock down lethality and general drug combination toxicity.

-Use of phosphoproteomics is unclear since erlotinib treatment and filtering criteria are not discussed in main text.

-Combination therapy experiments were focused on only one mechanism of EGFR TKI resistance (T790M mutation) and tested only with a single cell line.

By using a TAP-MS approach, the authors were able to generate a mutant EGFR network with protein-protein interactions specific to lung cancer cells. Additionally, by using a large-scale siRNA screen to determine the effect of specific proteins on cell viability, they were able to identify a core network downstream proteins necessary for mutant EGFR-mediated signaling. This yielded both specific drug targets and functional clusters. Cluster analysis was able to identify a specific protein that initially did not show a large effect on cell viability in the siRNA screen, but was later demonstrated to reduce cell viability to around 20% in both EGFR TKI sensitive and resistant cell lines. The most dramatic results in this paper were displayed through the use combining an irreversible EGFR TKI with drugs that target components of the EGFR mutant core network. Combination therapy resulted in a reduction in viability of EGFR TKI resistant cells to nearly zero when CDK9 inhibitors (AT-7519, BMS387032, or flavopiridol) were combined with afatinib. Furthermore, by focusing on established kinase inhibitors undergoing study in current clinical trials, these results could quickly translate into clinical testing.

Although TAP-MS generated a mutant EGFR lung cancer specific network, certain details about the network are unclear. The authors need to state how erlotinib treatment was incorporated into the phosphoproteomics data, as well as how the clusters are defined. Also, additional support for the use of the interactome data to determine novel drug targets would provide further validity for their methods. Since some of the identified targets are not specific to EGFR signaling (ex CDC37 cluster), they might have also identified these proteins through only an siRNA screen. Additionally, although the authors show dramatic results using combination therapy to target EGFR TKI resistant cells, these results were only demonstrated in one EGFR mutant cell line and were not tested in other contexts (further discussed below). Further validation using additional cell lines including those with different mutations that lead to EGFR TKI resistance (e.g. MET overexpression) should be done (detailed below).

Major comments

Two components of the identified EGFR core network were pursued further: "ARHG5 and CDC37 were further validated as key proteins in EGFR-mediated survival." The focus on the CDC37 chaperone protein was somewhat arbitrary since as acknowledged by the authors, knock down of CDC37 had less of an effect on viability than generally required by their screening thresholds. The specificity of the relationship between CDC37 protein and the EGFR signaling network to cellular viability was not adequately investigated. It remains possible that CDC37 knockdown would impact cell viability even in cells not dependent on mutant EGFR signaling - the effects were tested and seen in the EGFR E746-A750 and the E746-A750 + T790M signaling dependence contexts, but not in other i) EGFR, ii) RTK, or iii) non-RTK contexts. These issues need to be appropriately addressed.

For ARHG5, the presented EGFR TKI data (Fig 4D) does support a relationship between the EGFR signaling network and ARHG5, but for the viability data in Fig 4E there is again no investigation of other signaling dependency contexts (as described in i - iii above for CDC37). The statements "We observed no direct changes in either total EGFR or tyrosine phosphorylated EGFR (Figure 4F). Similarly, we observed no changes in downstream ERK or AKT phosphorylation..." do not match the data in Fig 4F.

The statement "Surprisingly, we identified midostaurin (PKC412) and lestaurtinib (CEP-701) as inhibitors with direct binding affinity to EGFR T790M alleles, which we further validated." is not clear (nor what is meant by 'surprisingly'). What is the source of the initial (pre-validation) identification? What validation experiments were done? Does this come directly from a reference or database - if so, this is not so much an identification by the authors as it is a literature-based fact. If so, provide the relevant reference(s) in this sentence. Upon knowledge of this fact, the subsequent experiments to combine these T790M EGFR inhibitors with erlotinib (to which the parental PC9 cells are sensitive) to treat the PC9GR cells (which have gained resistance to gefitinib via acquiring the T790M mutation) is rational even without any of the manuscript's preceding data. In other words, this co-treatment approach is not informed by the manuscript's proteomics-derived EGFR signaling network. Yet, this result falls within the manuscript subsection subtitled "Therapeutic opportunities identified through core network-directed chemo-informatics."

The EGFR inhibitor and CDK9/11A/11B (and PIK3CA) cotargeting approaches of Fig 7F are legitimately informed by the proteomics-derived EGFR signaling network data. As with the CDC37 and ARHG results, the CDK9/11A/11B studies lack proper controls to tie this co-targeting approach to the mutant EGFR signaling dependency context. As background, the authors should include whether mutant EGFR cells are more sensitive than non-mutant cells to the irreversible EGFR inhibitor afatinib used here. Then appropriate controls should be included in this regard for the co-treatment approach - specifically effects on the parental/sensitive cells, and effects on non-EGFR mutant cells. Would this co-targeting approach also work (not work) in other EGFR TKI resistance contexts (MET upregulation, EMT transition)? The CDK9/11A/11B co-targeting result should be mentioned in the abstract.

All combination treatments should include appropriate measures of additivity/synergy.

P. 7 In what sense does retroviral expression 'avoid highly overexpressed bait proteins'. No data of how endogenous and engineered expression of the bait proteins compares is shown. Provide data, or

provide references if sufficient.

P. 11 "we identified 3 functional clusters, in addition to EGFR, that strongly affected cellular proliferation in these EGFR-addicted lung cancer cells": It is unclear, how the clusters were defined and the clustering is not apparent in the network representation. Provide additional evidence for the existence of these three function clusters, and clearly indicate their definitions/boundaries in Fig 3C.

P. 17 "Combining afatinib with inhibitors of CDK9 proteins...": The best combination effects are achieved with CDK9 inhibitors. However, CDK9 entered the picture indirectly via its CDC37 interaction, which has a general, not EGFR-specific role as a co-chaperone. Considering this, discuss more critically what the contribution of the interactome data for the identification of the co-treatment targets is.

Combination therapy experiments: validate with additional TKI resistant cell lines. For Figure 7D/E/F, use an additional cell line with the T790M mutation. In addition, for Figure 7F, validate with HCC827ER cell line.

P. 9 "Using protein-protein interaction databases, we linked pTyr containing proteins to proteins identified in our TAP-MS experiments." : explain in more detail, how this was done, e.g. which databases, which filtering criteria. Were only pTyr sites that responded to Erlotinib treatment considered? Main text does not state that the cells used for phosphotyrosine proteomics were treated with erlotinib - please clarify. The supplemental methods specify criteria for determining phosphorylation sites that are significantly modulated by erlotinib and that only significant sites were added to the network. However, the figure 2 legend states phosphoproteins perturbed by erlotinib are indicated by pink ellipses, which makes it seem that both perturbed and unperturbed phosphoproteins were included in the network. Please state more clearly the criteria used to add phosphoproteins to the network. The supplemental data related to the pY results cannot be read as submitted. It is generally unclear how much the pY results added to the findings.

P. 10 "Multiple members of the EGFR complex were co-purified with ERBB2, including EGFR, ..., resulting in our finding that ERBB2 could potentially play an important role in modulating EGFR signaling.": With the strong interaction between EGFR and ERBB2 it is likely that a subset of the additional complex components are co-purified indirectly via EGFR (i.e. do not interact directly with ERBB2). With this, the authors need to be careful with conclusions based on "multiple" (direct and indirect) copurified complex members.

P. 13 "The EGFR TKI resistant cells were transfected with individual siRNAs of the entire library": The data is lacking from the supplement. For full reviewer and general reader evaluation it will be critical to include this data.

Figure 5: Include names and numbers also for the targets that are not shared between the parental and resistant cells. Clearly indicate the criteria used to determine if knock down of a gene target affects viability and to create the Venn diagrams.

P. 16 "We arrived at a drug network consisting of 26 compounds": The authors say the reduce the size of the original 1520 compounds by focusing on kinase inhibitors (this intermediate number should be indicated), but it is unclear how this number is further reduced to 26.

Does the overlap approach of Fig. 5 adequately take into account proteins important for viability of only EGFR TKI resistant cells? Discuss. Such proteins could provide additional potential drug targets.

Many figures are difficult to read, for example due to small fonts.

Minor comments

P. 6 "Mutant forms [plural] of EGFR were used as bait for TAP-MS": From the methods section it appears only one type of mutant was used (E746-A750). Please clarify. It would also be beneficial to add more background about the E746-A750 and L858R EGFR activating mutations (including what is indicated by the E746-A750' EGFR mutation notation) in the introduction to provide

rationale for generating interactomes for the mutants.

p. 7 "found in the GFP pull-down": clarify, whether a single GFP pull-down or an aggregation of several pull-down attempts was used

p. 9 "miss transient yet important interactions (such as pTyr-SH2)": Clarify, how "transient" interactions are defined in this context. Are you referring to a high k_{off} rate for these pTyr-SH2 interactions or to the transient nature of protein phosphorylation (which is less true in the presence of phosphatase inhibitors in the lysis buffer)? If the former, provide references for support.

P. 10 "ERBB3 bound EGFR but not ERBB2; therefore, EGFR may be the primary partner". What are the relative expression levels of the EGFR-family members in these cells? The ERBB2 interaction could have gone unnoticed, due to its lower expression.

P. 11 "Knockdown of other proteins, such as ARHG5, appeared to affect other pathways." It is later mentioned that ARHG5 is an activator of the Rho family of GTPases. To demonstrate ARHG5 is actually affecting other pathways, additional downstream targets should be tested.

P. 11 "Although cell viability for CDC37 was only 54%, ..." should read 'for CDC37 knock down ...'

P. 12 "ARHG5, which was identified as a novel core network protein affecting sensitivity of all mutant EGFR cell lines" should read 'affecting cell viability'.

P. 15 "(i) in vitro competition assays using purified kinase domains and inhibitors" should read 'and kinase inhibitors'.

P. 15 BindingDB and Drug Bank should be briefly described.

P. 16 "... were recovered equally" should be clarified along the lines of '... were recovered equally from cells with or without the T790M mutation.'

P. 38 Figure 2 legend "Tyrosine phosphorylated proteins perturbed by erlotinib are shown as pink ellipses, including EGFR, SHC1, and ERBB3." Unable to distinguish pink from red that is used to label proteins identified by TAP and phosphotyrosine proteomics.

Figure 6A and 6B display almost the same information. The only thing that is added by including 6A is that it shows some drugs target more than one protein in the core network.

Figure 7D and E: Previous experiments also looked at pAkt in addition to pEGFR and pErk. Is there a reason this was omitted for these experiments?

SI information, p. 3 "Tandem affinity purification (TAP) was performed as previously described(31).": provide full protocol.

SI information, p.3 "We also accepted single peptide hits, if the score was above a more stringent threshold": what was this threshold?

Submit supplemental data files as excel files, not PDF files.

Fig 3C. Y-axis labels incorrectly refer to phospho-protein levels as 'expression' levels.

Reviewer #3:

The manuscript develops a functional protein interactome for identification of vulnerable features of lung tumors driven by EGF receptor mutations, and in the context of common resistance mutation T790M. They begin by pulling down TAP-tagged EGFR activated mutants and ERBB3 expressed in two sensitive lung carcinoma cell lines. Binding partners revealed by MS/MS were evaluated

computationally to develop a network, and then a core subset was tagged and used for a second round of MS/MS to extend the interactome. The resulting data were evaluated together with complementary information on recovery of PTyr phosphopeptides from PC9 cells, and interpreted in the context of known binding partners for these phosphoproteins. These data were subject to DAVID analysis to identify the subset important for signaling-related pathways that were developed into a candidate target set of 102 target genes. Next, siRNAs were deployed against each of these candidate genes to determine the impact on cell growth. A core subset of 14 especially sensitive genes was identified. For EGFR-active lung cancer cell lines, erlotinib resistance over time is a major treatment issue. siRNAs were screened against lines with T790M and with resistance through other mechanisms. Both CDC37 and the guanine nucleotide exchange factor ARHG5 were analyzed in greater depth for impact on signaling and apoptosis in knockdown experiments. Finally, databases were searched to identify agents potentially targeting the EGFR-sensitizing targets. Two kinase inhibitors, midostaurine and CEP-701 paired well with EGFR inhibitor erlotinib to inhibit growth of cells with the EGFR T790M TKI-resistance mutation, and also combinations with the irreversible EGFR inhibitor afatinib.

These studies take a biologically well-justified hierarchical approach involving protein interactions, phosphoproteomics, and functional siRNA screening in an effort to identify important nodes in EGFR signaling pathways, and then apply this information to identification of candidate agents. Most of the experimental work is complete and well-documented, with some practical issues for the knockdown and drug treatment experiments. The most general issue is that the siRNA screens and drug testing do not address impact of the knockdowns and drug treatments on growth of cells without driver EGFR mutations. Many of the pathway targets are common among growth factor receptors, so the same components may be important for serum-driven cell proliferation in culture, which would affect the siRNA screens. A second issue, is that broad inhibitors such as staurosporine derivatives may hit such a broad array of targets as to be toxic to all cells. It's difficult to know whether there would be a meaningful therapeutic index without extending the drug treatment experiments to cells with other drivers and also to normal cells. Finally, it is not clear that either formal synergy (isobologram analysis) or apoptosis of a majority of the cells is achieved with the combinations. Extending the biological analyses would greatly strengthen the manuscript.

Other major points.

1. The DAVID analysis may be unnecessarily restrictive: should impact on metabolic genes be conserved? On protein translation?
2. Fig. 4: how do knockdowns affect normal cells and cancer cells without EGFR mutations? CDC37 may be essential for growth/viability of all cells? This is partly, but not entirely addressed in Fig. 5.
3. Fig. 7D: Only 50% growth inhibition- does this increase over longer time points? Is it specific to cells with the EGFR mutations?
4. Fig. 7E: Greater growth inhibition is more convincing for this combination, but is this specific to EGFR-mutant cells or cancer cells?
5. 7F: Here, much greater effect on the combinations, but again the question about toxicity for all cells vs. tumor cell selectivity. Caspase increment is very clear. But, what percentage of cells actually undergo apoptosis? This is best addressed by flow cytometry or other cell-based assay.

Other issues.

1. Need higher resolution graphics for the network diagrams: for example, in Fig. 2 phosphorylation site names are illegible and pink color is hard to pick out.
2. Not clear what is meant by the following sentence. "we manually examined all reported interactions between compounds binding EGFR proteins harboring the T790M allele" Were the compounds of interest those known to bind this allele, or what, and what is meant by manual examination?
3. ...with ARHG5 siRNA... "We observed no direct changes in either total EGFR or tyr phosphorylated EGFR (Fig. 4F)... changes in downstream ERK or AKT phosphorylation..." These results are variable between the two oligos, and between PC9 and PC9GR. For example, PEGFR is reduced with A4 in PC9, as does EGFR. pAKT goes up with A3.

Thank you for conveying the reviewer's comments on our manuscript. In response to the helpful comments, we have added additional experimental data and made major changes to help clarify the approaches, results, and conclusions.

First, one major concern was the lack of detailed data on the conserved vulnerability of the core EGFR network across additional lung cancer cell lines. To address this important point, we examined an additional 11 lung cancer cell lines, 10 of which do not depend on EGFR for survival, using the 14 core proteins identified in our proteomics analysis plus CDC37, with a new focused siRNA analysis on cell viability. Of the 15 siRNA examined (14 core network plus CDC37), 9 siRNA had significant differences in cell viability in the EGFR mutated cells compared to wild-type EGFR cells. This included EGFR, GRB2, MK12, SHC1, ARAF, CD11B, ARHG5, GLU2B, and CD11A. In addition to allowing nomination of EGFR mutant-specific targets, this allowed subsequent experiments to show that proteins with generalized toxicity across multiple cell lines predict generalized toxicity of drugs that target these proteins. In other words, interactome proteins surrounding mutant EGFR could be further defined and we could identify proteins whose inhibition results in generalized toxicity. For example, siRNA targeting the proteins affecting cyclin-dependent kinases (CDK9) had generalized effects across all cell lines examined (N=17), which agreed with effects of kinase inhibitors affecting these same proteins. We feel this is a valuable contribution by showing how interactome analysis followed by functional analysis can help better define subtype-specific targets as opposed to generalized targets. As such, we have modified the manuscript to reflect these new data and findings, especially as this argues against co-targeting proteins with generalized toxicity. We believe this could be a good approach to excluding compounds or drugs for further evaluation based on excessive toxicity or lack of specificity for a particular cancer subtype.

We also performed additional experiments to better define interacting proteins related to driver EGFR as opposed to wild-type EGFR. We performed additional TAP and mass spectrometry experiments using immortalized epithelial airway cells to compare EGFR binding proteins between mutant and wild-type forms of EGFR. We hypothesized that this additional approach would provide additional rationale for selecting secondary baits for iterative TAP experiments. While not explicitly asked for by the reviewers, we felt that these studies could provide additional insight into mutant EGFR interactomes. A new section in the results discusses these new data and how they relate to the studies in lung cancer cell lines. This also serves as a potential approach to functionalize the emerging mutational data from genomics studies by viewing mutational effects on protein interactions. This is also now discussed further.

Second, we have better clarified the phosphoproteomics experiments and reanalyzed our data to show how these data contributed to the final results.

Third, we have produced new data using additional lung cancer cell lines engineered to have T790M EGFR alleles to show combination effects of EGFR TKI with other compounds. At the same time, we have removed weaker data using drugs hitting proteins showing generalized toxicity as discussed above.

Finally, we have revisited all figures and have made improvements in clarity. At the same time, we realize that some of these figures may suffer in print form yet will be highly informative with electronic viewing.

Below, we address each reviewer's questions and critiques point-by-point. In italics, we copied the original reviewers' comments, and in bold we have stated our answers.

We are confident that the new version of the manuscript is now ready for publication and further highlights a systems medicine approach to targeting aberrant signaling interactomes in cancer. It is understandable that papers that start with systems-level approaches to analyze and understand systems move on with more focused validations. This is inevitable, and the only way to become concrete on exemplifying utility of the data and approach. Of course, given additional years of work, one could explore all possible drug combinations. However, we report on a success story and are

sure that the reader will value the effort taken in going from a global survey to a focused drug combination. We would like to thank again for your and the reviewers' consideration of our revised manuscript.

Reviewer #1:

The authors constructed a comprehensive interaction network of EGFR mutant and used it to explore drug resistance mechanisms and therapeutic opportunity. To this end, they performed tandem affinity purification-liquid chromatograph-mass spectrometry (TAP-LC-MS/MS). They then determined EGFR core networks that catalog proteins that highly affected the cell viability of cancer by using RNA interference screening. By profiling the perturbation effects of the core network and comparing them between drug-resistant cancer cell lines, they showed a strategy to elucidate drug resistance mechanisms. Finally they suggested a few drug combinations for potential cancer therapy. I have some points that the authors should address.

1: *In the abstract and page 11, the authors mentioned three modules (functional clusters) referring Figure 3C. However it is not recognizable from Figure 3C. The authors should clearly state how they recognized three distinct functional clusters.*

Response: We simply visualized what we called clusters. We realize that this may be confusing; thus we have reworded this sentence to state more clearly what we visualized.

2: *On page 8, the authors chose GRB2, SHC1, ERRFI, UBS3B, ERBB2, GRP78, AP2M1, and CDC37 for the second round of TAP based on their roles in linking EGFR to downstream MAPK signaling. This selection strategy can cause bias to the EGFR interactome networks and subsequent analysis. All the proteins from the first round of TAP should be used as bait in the second round of TAP. At least they should be chosen in an unbiased way.*

Response: We have performed additional experiments to better justify this approach. We used TAP to define EGFR interactomes in mutated EGFR directly compared to wild-type EGFR in immortalized airway epithelial cells (AALE). From this analysis, we could define mutant-specific interacting proteins, wild-type specific proteins, and shared proteins. These new data are shown in Figure 2B. We compared this list to the proteins found in the lung cancer cell lines. Importantly, a number of secondary baits are mutant EGFR specific (4 of the 8), 1 is shared between mutant and wild-type proteins, 2 are found only in wild-type proteins, and 1 was not identified in this experiment. Based on this, we feel our approach is reasonable.

Secondary Baits	AALE _TAP	
	Mutant EGFR	Wildtype EGFR
GRB2	Y	N
SHC1	Y	N
ERRFI	Y	N
UBS3B	Y	N
GRP78	Y	Y
ERBB2	N	Y
AP2M1	N	Y
CDC37	N	N

3: *On page 9, it should be described more clearly how the phosphotyrosine proteomics data were integrated. Even in the supplementary method section, it was not described in enough detail. It should be documented whether tyrosine kinase and SH2 domains were considered and how their specificities were considered when the authors added interactions.*

Response: We thank the reviewer for pointing out this deficiency and have now expanded the phosphoproteomics analysis, including more details on page 11 and a new figure (Supplemental Figure 7) on how this contributes to the interactome.

4: *The authors performed tyrosine phosphoproteomics only in PC9 cells. They should justify EGFR-driven phosphorylation signaling is not different between PC9 and the other cells the authors used, such as HCC827 cells. Otherwise, they also should perform phosphoproteomics in other cells.*

Response: This is now better explained in the text. Our previous study using purified SH2 domains to characterize phosphotyrosine signaling in lung cancer cells found PC9 clustering among other EGFR mutant cells (Machida, PLoS One, 2010). Similar results were found in a previous study using anti-phosphotyrosine antibodies and mass spectrometry (Rikova, Cell, 2007). Based on these results, we felt single analysis was a reasonable approach, especially as we were sampling to gain additional proteins to insert into the interactome and large number were already identified using TAP. This is now better explained in the text on page 11.

5: *On pages 11 and 12, the authors should provide a more reasonable justification of why they picked up CDC37 and ARHG5. For CDC37, they should provide logical ground or evidence when they say "Although cell viability for CDC37 was only 54%, its association with a module of protein kinases driving cell viability identified in the core network suggested an important functional role."*

Response: We have clarified that rationale for studies on CDC37 on page 17 and ARHG5 on page 18. CDC37 was chosen based on (i) its function as a co-chaperone with HSP90, (ii) known role of HSP90 in chaperone of mutant forms of receptor tyrosine kinases (*EGFR*, *EML4-ALK*), and (iii) emerging interest in CDC37 inhibitors for treatment of cancer. ARHG5 was a novel protein found in the EGFR interactome and siRNA-mediated knockdown showed more vulnerability in EGFR mutant lines than in wild-type EGFR lines (new Figure 5). Furthermore, the role of guanine exchange factor signaling in the context of mutant EGFR (or wild-type EGFR signaling for that matter) is poorly understood. These points are now added to text to help clarify the rationale for further studies on these proteins.

6: *On page 10, the authors used DAVID to identify proteins annotated to signal transduction pathways (*ERB*, *MAPK*, apoptosis, and cell cycle). The selected genes were subjected to the siRNA screening to determine the "core network". Using gene annotations to select candidate genes is highly risky since it causes biases to the subsequent analysis and prevents novel findings. The authors should use all the 265 genes for the siRNA screening to avoid potential bias or at least select candidate genes in an unbiased way.*

Response: We appreciate this concern. However, an alternative viewpoint is that filtering large system level proteomics datasets may enable better annotation of biologically relevant proteins/pathways in a more efficient process. Indeed, our results support this contention, as we could identify proteins that play integral roles in mutant EGFR signaling and could also define proteins with more generalized effects. Filtering also reduces the number of siRNA for study, which has been argued recently as a better strategy for use of RNA interference functional based assays (Kaelin, Use and Abuse of RNAi to Study Mammalian Gene Function, Science, 2012). In addition, we re-analyzed the 169 excluded proteins not chosen for secondary baits. Many of these appear as frequent hitters in TAP datasets (histone proteins, tubulins, heat shock proteins, chaperones, riboproteins); a DAVID analysis suggested these proteins are more involved in infectious/inflammatory signaling. We believe it is understandable that papers that start with systems-level approaches to analyze and understand systems move on with more focused validations. This is inevitable, and the only way to become concrete on exemplifying utility of the data and approach. Of course, given additional years of work, one could explore all possible targets in greater detail. However, we report on a success story and are sure that the reader will value the effort taken in going from a global survey to a focused functional analysis.

7: *On the sentence in page 12 "Knockdown of CDC37 reduced levels of EGFR and downstream pERK and pAKT, consistent with its role as a co-chaperone of EGFR with HSP90", did the author mean pEGFR instead of EGFR?*

Response: No, we meant to say EGFR, as mutant forms of EGFR are known client proteins of HSP90 and we hypothesized that CDC37 interference would affect this chaperone process and result in reduced levels of total EGFR protein.

8: *On page 15, the authors should state how they integrated in vitro kinase assays (Anastassiadis et al, 2011), since it does not provide kD values. Also these types of data are highly erroneous, a benchmark needs to be provided.*

Response: To appropriately take advantages of the Anastassiadis screening data, as it is relatively noisy, we have implemented a network-based method to derive more robust ranking scores. We have first constructed an "EGFR-specific" network based on our TAP experiments, in which the "EGFR-specific" interactions among the profiled kinases as potential drug targets were obtained. Based on the drug screening values from Anastassiadis et al., all potential drug candidates are connected to different kinases in this EGFR-specific network. The final ranking scores were derived by diffusing the given screening values in the Anastassiadis data based on the shorted distances between potential drug candidates to different kinases. The underlying assumption is that the drug candidates can be more effective for a specific kinase if they have strong effects on its close neighbors in this "EGFR-specific" network. Based on these derived ranking scores, we select a cutoff threshold values to derive the final lists of candidate kinase targets and corresponding drug candidates. These results contributed to results shown in Figure 6A, listing all possible compounds and targets within the core network.

We agree that these data can be prone to error. Therefore, we focused on higher quality data with Kd values to focus the compounds and, more importantly, further validated effects of midostaurin and lestaurtinib using in vitro kinase assays, chemical pulldowns, and Western analysis.

9: *On page 17, the authors should justify why they chose AT-7519, BMS387032, flavopiridol, and BKM120.*

Response: These drugs targeted proteins found in the EGFR core network (CD11A/B, CDK9). However, new analyses demonstrate that CDK9 loss has effects across all the cell types independent of EGFR mutation status (new Figure 5). We show now that, as predicted by the lack of specificity of CDK9 RNAi, these compounds have no clear cut difference in sensitivity between the EGFR mutant and wild-type cell lines (Supplemental Figure 6). Similarly, loss of PI3K subunits (p55G, p85B, PK3CA) similarly had effects not specific for EGFR mutant cells. Therefore, we removed studies on BKM120.

10: *Throughout the manuscript, there are a number of unclear sentences, which need to be revised or explained in more detail.*

a. *On page 12, "ARHG5, which was identified as a novel core network protein affecting sensitivity of all tested mutant EGFR cell lines"*

b. *On page 14, "common proteins used by EGFR (including T790M EGFR) in resistant cells may be considered targets to pursue in overcoming resistance to EGFR TKI, such as ARHG5 and CDC37, as we have shown above"*

c. *On page 17, "we examined efficacy of compounds reported to affect survival kinases located within the core network in EGFR TKI-resistant cells harboring T790M"*

Response: Thank you for pointing these out. We have expanded and/or reworded these sentences to increase clarity.

Minor Comments:

1) *Mixed use of 'Figure #' and 'Fig. #' throughout the manuscript.*

2) *When the authors say "the degree of overlap was reduced with loss of EGFR, GRB2, and SHC1 protein dependence" (page 14), it seems they omitted P55G.*

3) *On page 17, kinase-inhibitor interactions were stated in a confusing way.*

4) *On Figure 3C, the meaning of the gray boxes are unclear.*

5) *On Figure 4, it would be helpful to state the meaning of 'SC', 'G1', and 'F12'.*

6) On Figure 5, it would be more comprehensible if the two 'ER's are replaced by full names ('HC827ER' and 'HCC4006ER').

Response: These have been addressed; thank you for alerting us to these issues.

Reviewer #2:

Li et al. conduct an integrative analysis of the EGFR protein interaction network. They identify a core network of 14 proteins, which is critical to the survival of several EGFR-mutated lung cancer cells. Interestingly, cells with acquired resistance to EGFR inhibition demonstrated differential dependence on the core components. Integrating their interactome data with a drug target network the authors identify and validate (co-)targeting strategies for acquired EGFR inhibitor resistant cells.

Pros

*-The authors present a highly integrative analysis strategy combining interactome mapping, phosphoproteomics, and siRNA screens with targeted follow-up experiments.
-Combination therapy with drugs targeting the core network dramatically reduced cell viability of EGFR TKI resistant cells.*

Cons

*-The work lacks some important controls (e.g. in non-mutant EGFR lung cancer cells, and in normal-like cells) to test for general network component knock down lethality and general drug combination toxicity.
-Use of phosphoproteomics is unclear since erlotinib treatment and filtering criteria are not discussed in main text.
-Combination therapy experiments were focused on only one mechanism of EGFR TKI resistance (T790M mutation) and tested only with a single cell line.*

By using a TAP-MS approach, the authors were able to generate a mutant EGFR network with protein-protein interactions specific to lung cancer cells. Additionally, by using a large-scale siRNA screen to determine the effect of specific proteins on cell viability, they were able to identify a core network downstream proteins necessary for mutant EGFR-mediated signaling. This yielded both specific drug targets and functional clusters. Cluster analysis was able to identify a specific protein that initially did not show a large effect on cell viability in the siRNA screen, but was later demonstrated to reduce cell viability to around 20% in both EGFR TKI sensitive and resistant cell lines. The most dramatic results in this paper were displayed through the use combining an irreversible EGFR TKI with drugs that target components of the EGFR mutant core network. Combination therapy resulted in a reduction in viability of EGFR TKI resistant cells to nearly zero when CDK9 inhibitors (AT-7519, BMS387032, or flavopiridol) were combined with afatinib. Furthermore, by focusing on established kinase inhibitors undergoing study in current clinical trials, these results could quickly translate into clinical testing.

Although TAP-MS generated a mutant EGFR lung cancer specific network, certain details about the network are unclear. The authors need to state how erlotinib treatment was incorporated into the phosphoproteomics data, as well as how the clusters are defined. Also, additional support for the use of the interactome data to determine novel drug targets would provide further validity for their methods. Since some of the identified targets are not specific to EGFR signaling (ex CDC37 cluster), they might have also identified these proteins through only an siRNA screen. Additionally, although the authors show dramatic results using combination therapy to target EGFR TKI resistant cells, these results were only demonstrated in one EGFR mutant cell line and were not tested in other contexts (further discussed below). Further validation using additional cell lines including those with different mutations that lead to EGFR TKI resistance (e.g. MET overexpression) should be done (detailed below).

Response: We thank the reviewer for detailed critiques and recommendations. The reviewer appreciated the highly integrated analysis of interactome mapping and phosphoproteomic mapping using mass spectrometry followed by functional analysis with siRNA and targeted follow up or other validation experiments. We have addressed the major “cons” pointed out.

Specifically, we have expanded the siRNA analysis of the major proteins identified as core network hits in the EGFR mutant cells to an additional set of cell lines that lack mutant EGFR and are insensitive to EGFR TKI. These new data are now shown in Figure 5 and described on pages 16-18. We provide additional detailed explanation of the phosphoproteomics experiments on page 11, including a new figure to show how these data added to the TAP interactome data (Supplemental Figure 7). Finally, we expanded studies of combination drugs in additional cell lines by creating new engineered cells lines with T790M EGFR alleles to produce more resistant cell lines for additional experiments.

Major Comments:

Two components of the identified EGFR core network were pursued further: "ARHG5 and CDC37 were further validated as key proteins in EGFR-mediated survival." The focus on the CDC37 chaperone protein was somewhat arbitrary since as acknowledged by the authors, knock down of CDC37 had less of an effect on viability than generally required by their screening thresholds. The specificity of the relationship between CDC37 protein and the EGFR signaling network to cellular viability was not adequately investigated. It remains possible that CDC37 knockdown would impact cell viability even in cells not dependent on mutant EGFR signaling - the effects were tested and seen in the EGFR E746-A750 and the E746-A750 + T790M signaling dependence contexts, but not in other i) EGFR, ii) RTK, or iii) non-RTK contexts. These issues need to be appropriately addressed.

Response: We realize we did a poor job in explaining the rationale for further studies on CDC37. CDC37 was chosen based on (i) its function as a co-chaperone with HSP90, (ii) known role of HSP90 in chaperone of mutant forms of receptor tyrosine kinases (EGFR, EML4-ALK), and (iii) emerging interest in CDC37 inhibitors for treatment of cancer. With additional results of the siRNA screen in the non-mutant EGFR cells, we found more generalized effects of loss of CDC37. This may make some sense, as we found a number of kinases, including kinases involved in cell cycle progression, in complex with CDC37. However, based on the known function of HSP90 in aberrant RTK signaling, we examined this further to show, at least in these mutant EGFR cell lines, one anti-tumor mechanism is direct interference with mutant EGFR protein signaling. This we believe may aid further efforts at attacking mutant EGFR proteins using direct inhibitors of HSP90 and CDC37, depending on therapeutic windows in human clinical trials. We have expanded the text to reflect these findings and rationale for validation experiments.

For ARHG5, the presented EGFR TKI data (Fig 4D) does support a relationship between the EGFR signaling network and ARHG5, but for the viability data in Fig 4E there is again no investigation of other signaling dependency contexts (as described in i - iii above for CDC37). The statements "We observed no direct changes in either total EGFR or tyrosine phosphorylated EGFR (Figure 4F). Similarly, we observed no changes in downstream ERK or AKT phosphorylation..." do not match the data in Fig 4F.

Response: ARHG5 was a novel protein found in the EGFR interactome, and the role of guanine exchange factor signaling in the context of mutant EGFR (or wild-type EGFR signaling for that matter) is poorly understood. As stated above, additional experiments now have examined ARHG5 functional role across additional non-mutant EGFR lines and lines not sensitive to EGFR TKI. These new findings show that EGFR mutant and dependent lines are more vulnerable to ARHG5 loss compared with wild-type EGFR cells (new Figure 5). This protein is rather novel and thus deserves more study. However, we have been unable to discern a molecular mechanism despite multiple attempts with more experimentation. We examined changes in downstream signaling using protein arrays for signaling proteins and no obvious changes were observed. We cannot find consistent results suggesting alterations in EGFR or downstream MAPK or PI3K signaling.

The statement "Surprisingly, we identified midostaurin (PKC412) and lestaurtinib (CEP-701) as inhibitors with direct binding affinity to EGFR T790M alleles, which we further validated." is not clear (nor what is meant by 'surprisingly'). What is the source of the initial (pre-validation) identification? What validation experiments were done? Does this come directly from a reference or database - if so, this is not so much an identification by the authors as it is a literature-based fact. If

so, provide the relevant reference(s) in this sentence. Upon knowledge of this fact, the subsequent experiments to combine these T790M EGFR inhibitors with erlotinib (to which the parental PC9 cells are sensitive) to treat the PC9GR cells (which have gained resistance to gefitinib via acquiring the T790M mutation) is rational even without any of the manuscript's preceding data. In other words, this co-treatment approach is not informed by the manuscript's proteomics-derived EGFR signaling network. Yet, this result falls within the manuscript subsection subtitled "Therapeutic opportunities identified through core network-directed chemo-informatics."

Response: The main bulk of the studies consisted of the integrated proteomic studies coupled with functional analysis of proteins using siRNA and now expanded analysis across additional cell lines. At the end, we were curious to know if the results could be directly translated into actionable therapeutic strategies and therefore linked to drug:protein databases. It was through this strategy and approach that we found midostaurin and lestaurtinib as reported EGFR inhibitors. This was done through the described database searching. While it is true that the databases reflect literature, it can be very difficult to identify all relevant papers in the literature and mine datasets that are frequently found in the supplemental material. This, we would argue, is a strength of our approach, as we found these important interactions driven by the strategy of linking our analysis results to the database of drug targets. We were surprised that midostaurin had allele-specific EGFR targeting properties, since this drug has been promoted mainly as a FLT3 and PKC inhibitor. This prompted the chemical proteomics validation data and the additional validation data.

It is important to highlight the successes of this integrated approach starting with interactome characterization by mass spectrometry followed by RNAi analysis and linkage to drug target networks. We show the ability identify and therefore nominate targets that are necessary for mutant EGFR mediated survival signaling. This in turn lead to interrogation of drug databases that collated large-scale drug target datasets that can be difficult to mine and uncovered two compounds with activity against T790M EGFR isoforms. However, the approach does have limits, which we acknowledge, which could be improved on in future work. First, a number of nominated targets are not typical druggable targets with enzyme activity but rather consist of adaptor proteins lacking enzyme activity. Nonetheless, recent re-emerged attempts to drug the undruggable, including interfering with protein-protein interactions or covalently modifying proteins facilitating their degradation, should reprioritize these targets for future drug discovery efforts, as successful compounds will have molecular diseases to attack. Second, some of the nominated targets found in the functional analysis of the interactome have generalized effects across EGFR-dependent and independent lines. This could be a positive, however, as we can also therefore deprioritize targets that may lack therapeutic windows or have non-specific effects across multiple tumor types. Finally, lack of specificity of drugs or promiscuous that extends to non-specific targets also limits the approach. For example, compounds were identified that attacked specific proteins displaying vulnerability for EGFR mutant cells (CD11A/B) yet also have effects on proteins (CDK9) that have generalized anti-tumor effects across lung cancer cell lines. Certainly, the approach can work better with better drug:target datasets, as we used databases from the public domain which may lack more novel compounds or lack better specific compounds.

The EGFR inhibitor and CDK9/11A/11B (and PIK3CA) cotargeting approaches of Fig 7F are legitimately informed by the proteomics-derived EGFR signaling network data. As with the CDC37 and ARHG results, the CDK9/11A/11B studies lack proper controls to tie this co-targeting approach to the mutant EGFR signaling dependency context. As background, the authors should include whether mutant EGFR cells are more sensitive than non-mutant cells to the irreversible EGFR inhibitor afatinib used here. Then appropriate controls should be included in this regard for the co-treatment approach - specifically effects on the parental/sensitive cells, and effects on non-EGFR mutant cells. Would this co-targeting approach also work (not work) in other EGFR TKI resistance contexts (MET upregulation, EMT transition)? The CDK9/11A/11B co-targeting result should be mentioned in the abstract.

Response: As discussed above, the expanded analysis of key EGFR mutant sensitive proteins identified in the interactome analysis finds generalized toxicity of some of the proteins (CDK9 for example). CD11A/B however demonstrates vulnerability to siRNA more so in EGFR mutant cells; however, a search across compound:target databases could not identify

compounds specific to these proteins and without affects on other CDK, such as CDK9. Based on these results, we have removed afatinib studies, as we now show that the compounds hitting CD11A/B have non-specific effects across multiple cell types (Supplemental Figure 6). We believe this strategy however can help with target identification/validation unique or common across particular cancer subtypes.

All combination treatments should include appropriate measures of additivity/synergy.

Response: To measure of additivity /synergy in all combination treatment, we repeated all relevant experiments and calculated the combination index (CI) using CompuSyn software (<http://www.combosyn.com/>) and labeled CI in the revised Figure 6G and 6H.

P. 7 In what sense does retroviral expression 'avoid highly overexpressed bait proteins'. No data of how endogenous and engineered expression of the bait proteins compares is shown. Provide data, or provide references if sufficient.

Response: This was shown in our previous paper (Haura, Journal of Proteome Research, 2011), but we performed additional experiments with Western blotting in Supplemental Figure 1, allowing comparison of exogenous tagged bait proteins with endogenous proteins. We find equal amounts of exogenous tagged bait proteins when compared to endogenous proteins.

P. 11 "we identified 3 functional clusters, in addition to EGFR, that strongly affected cellular proliferation in these EGFR-addicted lung cancer cells": It is unclear, how the clusters were defined and the clustering is not apparent in the network representation. Provide additional evidence for the existence of these three function clusters, and clearly indicate their definitions/boundaries in Fig 3C.

Response: We simply visualized three groups that we called clusters. We realize this may be confusing; thus we have reworded this sentence to state more clearly what we visualized.

P. 17 "Combining afatinib with inhibitors of CDK9 proteins...": The best combination effects are achieved with CDK9 inhibitors. However, CDK9 entered the picture indirectly via its CDC37 interaction, which has a general, not EGFR-specific role as a co-chaperone. Considering this, discuss more critically what the contribution of the interactome data for the identification of the co-treatment targets is.

Response: The new analysis of additional cell lines demonstrates generalized effects of these CDC37 complexed proteins involved in cell cycle, and we have therefore modified text and removed data with these compounds.

P. 9 "Using protein-protein interaction databases, we linked pTyr containing proteins to proteins identified in our TAP-MS experiments." : explain in more detail, how this was done, e.g. which databases, which filtering criteria. Were only pTyr sites that responded to Erlotinib treatment considered? Main text does not state that the cells used for phosphotyrosine proteomics were treated with erlotinib - please clarify. The supplemental methods specify criteria for determining phosphorylation sites that are significantly modulated by erlotinib and that only significant sites were added to the network. However, the figure 2 legend states phosphoproteins perturbed by erlotinib are indicated by pink ellipses, which makes it seem that both perturbed and unperturbed phosphoproteins were included in the network. Please state more clearly the criteria used to add phosphoproteins to the network. The supplemental data related to the pY results cannot be read as submitted. It is generally unclear how much the pY results added to the findings.

Response: Details on how phosphoproteomic data were analyzed and added to TAP interactome results is now expanded on page 11. A total of 186 phosphotyrosine proteins were identified, of which 31 were added into the EGFR interactome via known protein-protein interactions, 19 were chosen for functional siRNA analysis, and 3 were identified as being critical for viability in EGFR mutant cell lines. A figure showing how the TAP and pY phosphoproteomics data added to the interactome through to the functional analysis is now shown in Supplemental Figure 7.

P. 10 "Multiple members of the EGFR complex were co-purified with ERBB2, including EGFR, ..., resulting in our finding that ERBB2 could potentially play an important role in modulating EGFR signaling.": With the strong interaction between EGFR and ERBB2 it is likely that a subset of the additional complex components are co-purified indirectly via EGFR (i.e. do not interact directly with ERBB2). With this, the authors need to be careful with conclusions based on "multiple" (direct and indirect) copurified complex members.

Response: We appreciate this point and have made clarifications to our data interpretation in the manuscript. We do not argue that the interactions are direct, as the TAP approach finds proteins in complex. We state in the text on page 13 "Multiple members of the EGFR complex were co-purified with ERBB2, including EGFR, GRB2, HS90A and HS90B, CDC37, ERRFI, and UBS3B (Supplementary Figure S1D), resulting in our finding that ERBB2 could potentially play an important role in modulating EGFR signaling"

P13: P. 13 "The EGFR TKI resistant cells were transfected with individual siRNAs of the entire library": The data is lacking from the supplement. For full reviewer and general reader evaluation it will be critical to include this data.

Response: This should have read "The EGFR TKI resistant cells were transfected with the pooled siRNAs of the entire library". This has been corrected. All siRNA data are included in Supplemental Data file S3.

Figure 5: Include names and numbers also for the targets that are not shared between the parental and resistant cells. Clearly indicate the criteria used to determine if knock down of a gene target affects viability and to create the Venn diagrams.

Response: Done. Thank you for this suggestion.

P. 16 "We arrived at a drug network consisting of 26 compounds": The authors say the reduce the size of the original 1520 compounds by focusing on kinase inhibitors (this intermediate number should be indicated), but it is unclear how this number is further reduced to 26.

Response: As we mentioned in the text, 1520 compounds are identified from four drug databases in our drug network. One of those drug databases, "in vitro competition assays using purified kinase domains and inhibitors" (Davis et al, 2011), focus on the kinase inhibitors which are approved drugs for clinic use or are in human clinical trials and provide the Kd values. We added additional text in this section to describe how this was done which can now be found on pages 20-21. From this analysis, we focused on 26 compounds for further study only based on this higher quality drug database.

Does the overlap approach of Fig. 5 adequately take into account proteins important for viability of only EGFR TKI resistant cells? Discuss. Such proteins could provide additional potential drug targets.

Response: This is a good point that we failed to recognize. We examined TKI resistant cells for siRNA that affected viability using our criteria. We found 6 siRNA having more effect in EGFR TKI resistant PC9GR and 3 siRNA having more effect in EGFR TKI resistant HCC827ER cells compared to EGFR TKI sensitive parental lines. No siRNA was found to significantly affect HCC4006ER cells. Only AP2M1 and 1433Z were in common between the two cell lines (PC9GR and HCC827ER). The magnitude of the effect was also rather modest. As AP2M1 is known to be involved in RTK endocytosis, the results warrant some additional follow up, but we have chosen not to pursue this at this time. We added this to the text on page 16, as it is likely some readers would have a similar question. The table below lists percent cellular viability for each target following siRNA treatment in the EGFR TKI sensitive and resistant paired cells (PC9 and HCC827).

siRNA	PC9	PC9GR	siRNA	HCC827	HCC827ER
AP2M1	61.1	48.1	AP2M1	57.1	33.0

POF1B	65.1	21.2	1433Z	78.5	46.7
NFH	62.6	24.9	STAT3	50.8	31.3
1433Z	55.2	25.6			
CLK3	81.2	45.3			
CDK1	62.3	46.8			

Many figures are difficult to read, for example due to small fonts.

Response: Figures have been reworked. In some cases, they will be difficult to view in print form but are best downloaded or viewed via the website.

Minor Comments:

P. 6 "Mutant forms [plural] of EGFR were used as bait for TAP-MS": From the methods section it appears only one type of mutant was used (E746-A750). Please clarify. It would also be beneficial to add more background about the E746-A750 and L858R EGFR activating mutations (including what is indicated by the E746-A750' EGFR mutation notation) in the introduction to provide rationale for generating interactomes for the mutants.

Response: These points have been clarified in Introduction and initial Results section.

p. 7 "found in the GFP pull-down": clarify, whether a single GFP pull-down or an aggregation of several pull-down attempts was used

Response: This was an aggregate of GFP pulldowns (N=4) and we have clarified this in the text

p. 9 "miss transient yet important interactions (such as pTyr-SH2)": Clarify, how "transient" interactions are defined in this context. Are you referring to a high k_{off} rate for these pTyr-SH2 interactions or to the transient nature of protein phosphorylation (which is less true in the presence of phosphatase inhibitors in the lysis buffer)? If the former, provide references for support.

Response: Transient is maybe not the best chosen description; what we meant to say is weaker interactions that may not survive the two step biochemical purification via TAP. As tyrosine phosphorylation allows protein complex formation through SH2 domains, we reasoned that adding additional proteins through tyrosine phosphoproteomics could expand the interactome. There are probably other approaches but this was one reasonable one that could be performed with the additional experimental phosphoproteomics data.

P. 10 "ERBB3 bound EGFR but not ERBB2; therefore, EGFR may be the primary partner". What are the relative expression levels of the EGFR-family members in these cells? The ERBB2 interaction could have gone unnoticed, due to its lower expression.

Response: Relative expression of all HER members in these cells is hard to gauge, as this would require a per protein accurate measurement using a technique such as multiple reaction monitoring. We acknowledge the reviewer's point that the interaction could still exist.

P. 11 "Knockdown of other proteins, such as ARHG5, appeared to affect other pathways." It is later mentioned that ARHG5 is an activator of the Rho family of GTPases. To demonstrate ARHG5 is actually affecting other pathways, additional downstream targets should be tested.

Response: We have examined how loss of ARHG5 affects downstream pathways by using protein arrays. This was unable to identify a clear cut downstream target. We also attempted examining Rho assays but these were inconclusive. The exact molecular mechanisms of how ARHG5 affects these cells are beyond the scope of the current manuscript, which is focused more on the system level interrogations, but need to be done.

P. 12 "ARHG5, which was identified as a novel core network protein affecting sensitivity of all mutant EGFR cell lines" should read 'affecting cell viability'.

Response: Corrected

P. 15 "(i) in vitro competition assays using purified kinase domains and inhibitors" should read 'and kinase inhibitors'.

Response: Corrected

P. 15 BindingDB and Drug Bank should be briefly described.

Response: We have added additional description of the drug databases on page 31-32, including appropriate references to aid the reader. Thank you for this suggestion.

P. 16 "... were recovered equally" should be clarified along the lines of '... were recovered equally from cells with or without the T790M mutation.'

Response: Corrected

P. 38 Figure 2 legend "Tyrosine phosphorylated proteins perturbed by erlotinib are shown as pink ellipses, including EGFR, SHC1, and ERBB3." Unable to distinguish pink from red that is used to label proteins identified by TAP and phosphotyrosine proteomics.

Response: Figures have been modified to allow better visualization.

Figure 6A and 6B display almost the same information. The only thing that is added by including 6A is that it shows some drugs target more than one protein in the core network.

Response: We have removed Figure 6B.

Figure 7D and E: Previous experiments also looked at pAkt in addition to pEGFR and pErk. Is there a reason this was omitted for these experiments?

Response: We performed in-cell Western analyses to look at pERK and serine phosphorylation on Akt as part of the initial analysis of the core network siRNA analysis in Figure 3D. We subsequently examined pAKT in follow on validation studies of CDC37 and ARHG5. We used pERK only in anti-EGFR TKI studies as part of Figure 6 since this in our hands is a highly robust downstream assessment of EGFR kinase activity in these EGFR mutated lung cancer lines.

SI information, p. 3 "Tandem affinity purification (TAP) was performed as previously described(31).": provide full protocol.

Response: Corrected. We put the full TAP protocol in Supplemental Materials & Methods on page 2.

SI information, p.3 "We also accepted single peptide hits, if the score was above a more stringent threshold": what was this threshold?

Response: Corrected. Found in in Supplemental Materials & Methods on page 2-3.

Submit supplemental data files as excel files, not PDF files.

Response: Corrected.

Fig 3C. Y-axis labels incorrectly refer to phospho-protein levels as 'expression' levels.

Response: Corrected.

Reviewer #3:

The manuscript develops a functional protein interactome for identification of vulnerable features of lung tumors driven by EGF receptor mutations, and in the context of common resistance mutation T790M. They begin by pulling down TAP-tagged EGFR activated mutants and ERBB3 expressed in two sensitive lung carcinoma cell lines. Binding partners revealed by MS/MS were evaluated computationally to develop a network, and then a core subset was tagged and used for a second round of MS/MS to extend the interactome. The resulting data were evaluated together with complementary information on recovery of PTyr phosphopeptides from PC9 cells, and interpreted in the context of known binding partners for these phosphoproteins. These data were subject to DAVID analysis to identify the subset important for signaling-related pathways that were developed into a candidate target set of 102 target genes. Next, siRNAs were deployed against each of these candidate genes to determine the impact on cell growth. A core subset of 14 especially sensitive genes was identified. For EGFR-active lung cancer cell lines, erlotinib resistance over time is a major treatment issue. siRNAs were screened against lines with T790M and with resistance through other mechanisms. Both CDC37 and the guanine nucleotide exchange factor ARHG5 were analyzed in greater depth for impact on signaling and apoptosis in knockdown experiments. Finally, databases were searched to identify agents potentially targeting the EGFR-sensitizing targets. Two kinase inhibitors, midostaurine and CEP-701 paired well with EGFR inhibitor erlotinib to inhibit growth of cells with the EGFR T790M TKI-resistance mutation, and also combinations with the irreversible EGFR inhibitor afatinib.

These studies take a biologically well-justified hierarchical approach involving protein interactions, phosphoproteomics, and functional siRNA screening in an effort to identify important nodes in EGFR signaling pathways, and then apply this information to identification of candidate agents. Most of the experimental work is complete and well-documented, with some practical issues for the knockdown and drug treatment experiments. The most general issue is that the siRNA screens and drug testing do not address impact of the knockdowns and drug treatments on growth of cells without driver EGFR mutations. Many of the pathway targets are common among growth factor receptors, so the same components may be important for serum-driven cell proliferation in culture, which would affect the siRNA screens. A second issue, is that broad inhibitors such as staurosporine derivatives may hit such a broad array of targets as to be toxic to all cells. It's difficult to know whether there would be a meaningful therapeutic index without extending the drug treatment experiments to cells with other drivers and also to normal cells. Finally, it is not clear that either formal synergy (isobologram analysis) or apoptosis of a majority of the cells is achieved with the combinations. Extending the biological analyses would greatly strengthen the manuscript.

Response: We have addressed the reviewer's three major concerns.

1: "The most general issue is that the siRNA screens and drug testing do not address impact of the knockdowns and drug treatments on growth of cells without driver EGFR mutations".

Response: To address this important point, we examined an additional 11 lung cancer cell lines, 10 of which do not depend on EGFR for survival, using the 14 core proteins identified in our proteomics analysis and focused siRNA analysis. Of the 15 siRNA examined (14 core network plus CDC37), 9 siRNA had significant differences in cell viability in the EGFR mutated cells compared to wildtype EGFR cells. This included EGFR, GRB2, MK12, SHC1, ARAF, CD11B, ARHG5, GLU2B, and CD11A. In addition to allowing nomination of EGFR-mutant specific targets, this allowed subsequent experiments to show that proteins with generalized toxicity across multiple cell lines predict generalized toxicity of drugs that target these proteins. These data are now shown in new Figure 5.

2: A second issue, is that broad inhibitors such as staurosporine derivatives may hit such a broad array of targets as to be toxic to all cells. It's difficult to know whether there would be a meaningful therapeutic index without extending the drug treatment experiments to cells with other drivers and also to normal cells.

Response: The new experiments expanding siRNA analysis to additional cell lines has helped clarify which targets are specific and which are not for EGFR mutated cells. Importantly, this allowed subsequent experiments to show that proteins with generalized toxicity across

multiple cell lines predict generalized toxicity of drugs that target these proteins. In other words, interactome proteins surrounding mutant EGFR could be further defined and we could identify proteins whose inhibition results in generalized toxicity. For example, siRNA targeting the proteins affecting cyclin dependent kinases (CDK9) had generalized effects across all cell lines examined (N=17), which agreed with effects of kinase inhibitors affecting these same proteins. We feel this is a valuable contribution by showing how interactome analysis followed by functional analysis can help better define subtype specific targets as opposed to generalized targets. As such, we have modified the manuscript to reflect these new data and findings, especially as this argues against co-targeting proteins with generalized toxicity. This we believe could be a good approach to excluding compounds or drugs for further evaluation based on excessive toxicity or lack of specificity for a particular cancer subtype. Judging the feasibility of therapeutic windows is, admittedly, probably beyond the scope of the paper. The most important point is the systems-level evaluation of a principle. We assume that therapeutic windows will greatly depend by the dosage ultimately required that will depend on too many parameters to be assessed here. Our results do show that combination effects, with more pronounced apoptosis and synergy, can be observed in EGFR mutant cell lines compared to wildtype cell lines, in new Figure 6. It is important to note that the staurosporine derivatives that we specifically studied appear well tolerated in human clinical trials.

3: *Finally, it is not clear that either formal synergy (isobologram analysis) or apoptosis of a majority of the cells is achieved with the combinations.*

Response: Combination studies with erlotinib and either midostaurin or lestaurtinib were repeated using two apoptosis assays (PARP cleavage by western blotting and cleaved-caspase-3 by flow cytometry) and formal synergy studies were performed and CI reported. These data are shown in Figure 6F, 6G, and 6H.

Other major points:

1: *The DAVID analysis may be unnecessarily restrictive: should impact on metabolic genes be conserved? On protein translation?*

Response: We appreciate this concern. However, an alternative viewpoint is that filtering large system level proteomics datasets may enable better annotation of biologically relevant proteins/pathways in a more efficient process. Indeed, our results support this contention, as we could identify proteins that play integral roles in mutant EGFR signaling and could also define proteins with more generalized effects. Filtering also reduces the number of siRNA for study, which has been argued in the literature as a better strategy for use of RNA interference functional based assays (Kaelin, Use and Abuse of RNAi to Study Mammalian Gene Function, Science, 2012).

2: *Fig. 4: how do knockdowns affect normal cells and cancer cells without EGFR mutations? CDC37 may be essential for growth/viability of all cells? This is partly, but not entirely addressed in Fig. 5.*

Response: This is now addressed – see answer to major concern #1 above.

3: *Fig. 7D: Only 50% growth inhibition- does this increase over longer time points? Is it specific to cells with the EGFR mutations?*

Response: We performed cell viability assay with titration dose of drugs for 5 day treatment and the inhibition of combination group increased to 80%. To test whether this combination effect is specific to cell with EGFR T790M mutant, we performed cell viability assay in three cell lines harboring activating EGFR mutations along with EGFR T790M and two cell lines with wildtype EGFR. The combination of midostaurin with erlotinib exhibits synergy across the three of EGFR T790M cell lines while not observed in the cell lines with wildtype EGFR. The combination also demonstrates increased apoptosis in EGFR mutant cells. The data shown in Figure 6 and suggested this combination effect is specific to cell with EGFR T790M mutant.

4: *.Fig. 7E: Greater growth inhibition is more convincing for this combination, but is this specific to EGFR-mutant cells or cancer cells?*

Response: To test whether this combination effect is specific to cell with EGFR T790M mutant, we performed cell viability assay in three of EGFR T790M cell lines and two of EGFR wild type cells. Combination CEP-701 with erlotinib does synergistically affect all three of EGFR T790M cell lines, while not so in the two EGFR wild type cell lines. Similar to results with midostaurin, we observed more apoptosis in mutant EGFR lines compared to wildtype EGFR lines. The data shown in Figure 6 suggested that this combination effect is specific to cells with EGFR T790M mutant.

5: *7F: Here, much greater effect on the combinations, but again the question about toxicity for all cells vs. tumor cell selectivity. Caspase increment is very clear. But, what percentage of cells actually undergo apoptosis? This is best addressed by flow cytometry or other cell-based assay.*

Response: These data using broad spectrum CDK inhibitors have been removed given the siRNA results in additional cell lines showed knockdown of CDK9 was toxic across the cells and the CDK inhibitors don't have specificity to the more specific targets CD11A/B. We show that lack of specificity of CDK9 translates to lack of specificity of the CDK9/CD11A/B targeting compounds in Supplemental Figure 6.

Other Issues:

1: *Need higher resolution graphics for the network diagrams: for example, in Fig. 2 phosphorylation site names are illegible and pink color is hard to pick out.*

Response: Figures are improved and high-resolution figures have been created.

2: *Not clear what is meant by the following sentence. "we manually examined all reported interactions between compounds binding EGFR proteins harboring the T790M allele" Were the compounds of interest those known to bind this allele, or what, and what is meant by manual examination?*

Response: Corrected. We search for compounds in the databases that have activity against T790M EGFR alleles.

3: *...with ARHG5 siRNA... "We observed no direct changes in either total EGFR or tyr phosphorylated EGFR (Fig. 4F)... changes in downstream ERK or AKT phosphorylation..." These results are variable between the two oligos, and between PC9 and PC9GR. For example, PEGFR is reduced with A4 in PC9, as does EGFR. pAKT goes up with A3.*

Response: We performed additional experiments to examine how loss of ARHG5 affects downstream pathways by using protein arrays and western blotting. This was unable to identify a clear cut downstream target or mechanism. We also attempted examining Rho assays but these were inconclusive. The exact molecular mechanisms of how ARHG5 affects these cells are beyond the scope of the current manuscript, which is focused more on the system level interrogations, but need to be done. These data have been removed from the current form of the manuscript.

Thank you again for submitting your work to Molecular Systems Biology. We have now heard back from the two referees who accepted to evaluate the revised study. As you will see, the referees acknowledge that the modifications included in this revision significantly improve the study. Reviewer #2 however still raises some concerns that should be convincingly addressed. The two

main points raised by this reviewer refer to the need to considerably tone down some of the claims and to present the findings in a more rigorous way. The recommendations provided by this reviewer are very clear in this regard and we would kindly ask you to carefully address them in a last and exceptional second round of revision with suitable amendments to the text.

On a more editorial level, we would ask you to address the following points:

- We would kindly ask you to submit the molecular interactions described in this study to an appropriate database from the IMEx consortium (www.imexconsortium.org) and indicate the respective accession numbers in Materials & Methods.
- Please submit the phosphoproteomic dataset to a suitable public database and include the respective accession number in Materials & Methods.
- We would be most grateful if you could submit as 'source data files' the Cytoscape files (or equivalent) for the networks shown in Figure 3A, so that readers can quickly download these files to the desktop application.
- It is possible that the images from some of the blots shown in this study have been assembled by splicing non-contiguous lanes together (for example: Fig. 5D). If this is the case, these images would not satisfy our standards in terms of data presentation (see <http://www.nature.com/msb/authors/index.html#a3.4.3> "Vertically sliced images that juxtapose lanes that were non-adjacent in the gel must have a clear separation or a black line delineating the boundary between the gels"). We would ask you to go over all the figures and double check that all the data are presented in a rigorous manner according to our guidelines. In particular, if non-adjacent lanes are shown, please indicate this clearly in the figure and in the figure legend as well with a short explanation.

Reviewer #2:

In this revised manuscript, the authors performed additional experiments to distinguish components of wild type and mutant EGFR-dependent signaling. Additional TAP coupled with LC-MS/MS experiments compared binding proteins of mutant and wild type EGFR. They expanded their examination of the 14 protein core network to include lung cancer cell lines without EGFR mutations. This analysis revealed EGFR mutant cell lines were more sensitive to siRNA targeting of 9 members of the core network, while targeting the remainder of the network reduced cell viability in both mutant and wild type EGFR cell lines. This addressed one of the major concerns that non-mutant EGFR lung cancer cells should be tested for core network knock down lethality. Similarly, drug combination experiments were also performed in additional lung cancer cell lines with both mutant and wild type EGFR, and furthermore include measures of synergy.

The additional experiments showed that drugs targeting some components of the core network also showed toxicity in lung cancer cell lines with wild type EGFR, and thus some combination studies with these compounds were removed in the revised manuscript. There is an increase in experimental results tied to targeting ARHG5, CDC37, CD11A/B, and CDK9, but overall they could be better integrated into the flow of the manuscript as the current presentation gives a somewhat fragmented feel (a specific example of this is listed below).

The authors expanded their discussion of their phosphoproteomics experiments, which clarified how these proteins were added to the network. Additionally, they targeted EGFR TKI resistant cells with additional siRNAs to determine if there are greater effects in resistant compared to parental cells. The authors demonstrated that mutant EGFR cell lines are more sensitive to ARHG5 targeting and performed a protein array to investigate downstream signaling components, which unfortunately did not yield any consistent results. Figures are now more consistent and easier to read, which allows for better visualization of the core network and clusters.

The authors have generated a mutant-EGFR interactome that provided potential drug targets. However, the most promising targets within the core network lacked available drugs (as in the case of ARHG5) or current drugs had "cross-target" toxic effects in wild type EGFR lung cancer lines (drugs that affect both CDK9 and CD11A/B). The impact of one of the more dramatic synergy results in the original submission had to be substantially revised from the subsequent submission because the previously missing control of testing in non-EGFR mutant cells demonstrated "that CDK9 loss has effects across all the cell types independent of EGFR mutation status".

Major points:

As indicated in the prior review, the author's continue to oversell the presentation of their "core network-directed chemo-informatics" approach and its claimed successes/predictions. This is seen the following two aspects:

The main combination therapy experiments demonstrating synergy with EGFR TKIs were done with compounds that binds T790M mutant EGFR. While combining erlotinib (EGFR inhibitor) with midostaurin or lestaurtinib (EGFR T790M inhibitors) provides an opportunity to overcome EGFR TKI resistance, the rationale for this combination does not require the data from the mutant-EGFR interactome, a point made in the original review. If one were to have started the project at the step in the manuscript marked by "Because some databases report allele-specific interactions of compounds, we additionally searched for reported interactions between compounds binding EGFR proteins harboring the T790M gatekeeper allele." (P.21), one could have accomplished the same discovery. It may be true that analysis of the core network results inspired this subsequent step and this finding, but the finding was not dependent on the prior interactome work and this aspect should be appropriately presented/acknowledged. In this regard, the rebuttal text that describes the overall discovery procedure is more detailed and representative than the corresponding text in the manuscript.

P.23: "Finally, we tested if predictions made by the functional interrogation...": In the previous iteration of the manuscript the same inhibitors (AT-7519 and BMS-387032) were presented as a good choice for mutant EGFR-targeted therapy. In this version, the inclusion of siRNA screening in wildtype EGFR cell lines suggested these targets would have generalized (non-EGFR mutant specific) effects. It is unclear (and might be misleading for the reader), whether the subsequently observed non-differential (correctly predicted) or inverse differential inhibitor effects (incorrectly predicted) were accurately predicted by an a priori hypothesis, ie whether the approach can reproducibly "make predictions about lack of sensitivity [which would more appropriately be called specificity]". Supp Fig S6 includes additional data for the CDK9 inhibitor flavopiridol that are not discussed in the main text (although mentioned in the text of the original submission). Results for the CDK9 inhibitor flavopiridol would be expected to be non-specific (based on the siRNA data for CDK9 that is discussed in the text). However, in Supp Fig S6 the results show that with statistical significance flavopiridol has a lower IC50 in the mutant EGFR group. In sum, three related predictions were made for non-differential effects, and two were incorrect (one involving an experimental result of wildtype specificity, and one of mutant specificity). Furthermore, no predictions for mutant-specific cases were made nor tested. Thus the ability for this approach to make accurate predictions is not supported, and accordingly the overall utility is not supported. This section should be removed..

Minor points:

Figure 3A legend: While the main text adequately explains how the pY profiling data is integrated into the network, the figure legend is slightly unclear. It currently reads, "Tyrosine phosphorylated proteins significantly perturbed by erlotinib identified from pY experiments, and are shown as green ellipses, including EGFR, SHC1, and ERBB3. Those green ellipses nodes were added based on the public interactions database. The identified phosphotyrosine sites (n = 62) are indicated as blue circles connected to the relevant protein." It would be clearer if the authors would rephrase to "Green ellipses indicate proteins identified from pY experiments that were added to the network based on the public interactions database. Phosphotyrosine sites significantly perturbed by erlotinib (n = 62) are indicated as blue circles connected to the relevant protein."

P. 16-17: Although the authors addressed the comment to explain their rationale for targeting

CDC37, this data seems out of place inserted into the paragraph that determines which of the 14 core network proteins are specific to mutant EGFR survival. It would improve the clarity of the manuscript if this section were better integrated into the text.

Figure 6F: It would provide additional clarity if it was mentioned in either the main text or the figure legend that H157 is a wild type EGFR lung cancer cell line. The text mentions these cell lines when referring to figure 6G and H, but not 6F.

P. 11: "the Pearson's chi-squared goodness-of-fit test P value was calculated to estimate...": Explain the rationale for this statistical filtering approach. Include detailed description of phospho-proteomics method and description of statistical filtering procedure in methods section. Without detailed information it is difficult to appreciate how the authors obtained information on both phospho- and non-phospho- peptides. For the chi-squared approach, how do you motivate/justify the multiplication of the p-values, since they appear highly dependent? Was a multiple hypotheses correction performed?

P. 22: Mention and refer to the publication by Lee et al (2013) on midostaurin as an EGFR T790M inhibitor in the 1st paragraph of this section. The reader will likely miss this additional support for this finding if this publication is only very briefly mentioned in the discussion section.

More detail should be added to support figure legends to reduce the need for the reader to refer back to the main text for aspects such as inhibitor targets.

Reviewer #3:

This is an innovative hierarchical analysis that reveals considerable new information on signaling pathways and vulnerabilities of mutant EGFR-driven pathways in lung adenocarcinoma. The authors have taken an intensive vertically integrated approach to hierarchical discovery of these weak points. Given the scope of the turf, there are many ways to approach this problem, and the approaches taken here are reasonable. The authors have responded with text changes and additional experimental work to my major concerns and, in my view, those of the other reviewers.

2nd Revision - authors' response

26 September 2013

Thank you for conveying the reviewer's additional comments on our manuscript. We appreciate the ability to address the recommendations in this exceptional second round of revision.

We have addressed your points on the editorial level:

1. Interactome data and phosphoproteomic data have been deposited to public databases, and the accession numbers are listed in Materials and Methods.
2. A Cytoscape file corresponding to Figure 3A has been included.
3. We confirm that images from Westerns have not been spliced together – this is an imaging artifact that we have attempted to correct by rescanning the images.

Response to Reviewers

In response to Reviewer #2, we have made two major changes in response to the issues raised under "Major Points" [*“As indicated in the prior review, the author's continue to oversell the presentation of their “core network-directed chemo-informatics” approach and its claimed successes/predictions. This is seen in the following two aspects . . . “*]

First, we modified the text, both in the Results and Discussion sections, acknowledging that the synergy experiments, while inspired by the interactome analysis, were not dependent on the data. These statements in the Discussion match our last rebuttal text that the reviewer felt better represented the overall discovery procedure, including the limit on the T790M mutant EGFR.

Results Section Edits:

- **Page 19: We revised the section title** from “Therapeutic opportunities identified through core network directed chemo-informatics” to “Chemical compounds targeting the core functional network” – we believe that this best describes the overall approach while toning down the absolute dependence of the results on the core network.
- **Page 21-22: We revised the paragraph to state upfront:** “We also re-identified midostaurin (PKC412) and lestaurtinib (CEP-701), two compounds with FL3 activity and being evaluated with leukemia, as inhibitors with direct binding affinity to EGFR T790M alleles (Lee et al, 2013). Based on these findings, and the lack of other more prominent and specific compound:target interactions, we examined these potential T790M EGFR inhibitors in more detail.”

Discussion Edits:

- **Page 25:** “At the end, we were curious to know if the results could be directly translated into actionable therapeutic strategies and therefore linked to drug:protein databases. It was through this strategy and approach that we found midostaurin and lestaurtinib as reported T790M EGFR inhibitors. This was done through the described database searching. While it is true that the databases reflect literature, it can be very difficult to identify all relevant papers in the literature and mine datasets that are frequently found the supplemental material. This, we would argue, is a strength of our approach, as we found these important interactions driven by the strategy of linking our analysis results to the database of drug targets. Nonetheless, we acknowledge that, while inspired by the interactome analysis, one could have accomplished the same discovery of the drug synergy by focusing initially on database searches of T790M EGFR interacting compounds. We were initially surprised that midostaurin had allele-specific EGFR targeting properties, since this drug has been promoted mainly as a FLT3 and PKC inhibitor. This prompted the chemical proteomics validation data and the additional validation data.”
- **Page 26: This section was removed:** “Our general results support the idea that interactomes linked to drug databases can exploit drug promiscuity, enabling re-purposing of compounds for rationale combination therapy directed against genotype-specific lung cancers. This can be applied to new genomic landscapes to enable drugging new drivers.”
- **Page 27:** “It is important to highlight the successes and limits of this integrated approach starting with interactome characterization by mass spectrometry followed by RNAi analysis and linkage to drug target networks. We show the ability identify and therefore nominate targets that are necessary for mutant EGFR mediated survival signaling. This in turn lead to interrogation of drug databases that collated large scale drug target datasets that can be difficult to mine and uncovered two compounds with activity against T790M EGFR isoforms. However, the approach does have limits, which we acknowledge, which could be improved on in future work. First, a number of nominated targets are not typical druggable targets with enzyme activity but rather consist of adaptor proteins lacking enzyme activity. Most of the key proteins had no reported compounds or had compounds not useful in clinical studies. For example, no reported compounds were identified that interfere with ARGH5, identified in our analysis as being especially vulnerable in EGFR mutated cells. Nonetheless, recent re-emerged attempts to drug the undruggable, including interfering with protein-protein interactions or covalently modifying proteins facilitating their degradation, should reprioritize these targets for future drug discovery efforts, as successful compounds will have molecular diseases to attack (Neklesa & Crews, 2012; Neklesa et al, 2011; Patgiri et al, 2011). Second, some of the nominated targets found in the functional analysis of the interactome have generalized effects across EGFR dependent and independent lines. This could be a positive however as we can also therefore deprioritize targets that may lack therapeutic windows or have non-specific effects across multiple tumor types. Finally, lack of specificity of drugs or promiscuous that extends to non-specific targets also limits the approach. For example, compounds were identified that attacked specific proteins displaying vulnerability for EGFR mutant cells (CD11A/B) yet also have effects on proteins (CDK9) that have generalized anti-tumor effects across lung cancer cell lines. Better results can be anticipated with increased high quality drug:target datasets, as we used databases from the public domain which may lack more novel

compounds or lack better specific compounds. Further examination of other drugs and targets identified through our analysis may reveal additional co-targeting strategies.”

Second, we removed the last paragraph and experimental data shown in Supplemental Figure S6, as suggested by the reviewer. We appreciate that these data were less developed and less rigorous than other sections of the manuscript and its departure has minimal effects; in fact, we agree this will be less confusing for the readers.

Minor points suggested by Reviewer2.

- Figure 3A legend: Thank you for this suggestion – we have modified the legend using this improved language.
- p. 16-17: The text has been modified to improve clarity. We moved the section describing CDC37 results up two pages, which now follows the results on the initial siRNA screen.
- Figure 6F: We modified the main text and legend to state that H157 is a wild-type EGFR cell line.
- p. 11: Detailed descriptions of the statistical methodology used to analyze the phosphoproteomics data were originally placed in the Supplementary Methods, but we have now moved this into the Methods to add clarity.

Regarding the rationale for the Pearson’s testing, the overall design was to add in additional phosphotyrosine data onto the TAP backbone, producing an EGFR interactome that could subsequently be functionally interrogated. Rather than placing all phosphotyrosine data into the network (which could have been one approach), we inserted only proteins where some evidence existed of altered levels following EGFR TKI.

Now the question is – among the phosphorylation sites that we have identified, how many are regulated upon treatment? We have to state that this is a very tricky question as both the protein levels and phosphorylation states could be modified at the same time. To confidently answer this question for every peptide, we need to have precise measurements of abundance for both protein and a given peptide in a specific modification states (e.g. iTRAQ experiments). We don’t have such data; for each peptide modification state, we only have a number of spectra that were identified as having or missing given modification. This is not a precise measure of abundance, but still it should capture the significant changes in phosphorylation regulation. Therefore, the multiplication of Pearson chi-squared based P values is just a simple and natural way to design a score to rank phosphorylation changes that we observe and discard obvious sites that are not affected by the treatment. However, we cannot claim any statistical significance since the data that we use for scoring is not precise. The reviewer is correct in that Fisher method of combining P values would not be statistically correct, but the calculation of real P values was not our goal here.

- p. 22: The Lee reference is now added in the beginning of the paragraph.
- “More detail should be added to sup figure legends” — Supplemental figure legends have been expanded and re-edited to improve clarity.

We are confident that the new version of the manuscript is now ready for publication and further highlights a systems medicine approach to targeting aberrant signaling interactomes in cancer. We would like to thank again for your and the reviewers’ consideration of our revised manuscript, especially allowing us the opportunity to address final concerns in a second round of revisions. The reviews have been quite helpful, and the commentary during this process that would accompany an accepted paper would go a long way further help the readers.